# Cystatin B is essential for proliferation and interneuron migration in individuals with EPM1 epilepsy

Francesco Di Matteo[1,2,†], Fabrizia Pipicelli[1,2,†], Christina Kyrousi[1], Isabella Tovecci[1,3], Eduardo Penna[3], Marianna Crispino[3], Angela Chambery[4], Rosita Russo[4], Ane Cristina Ayo-Martin[1,2], Martina Giordano[1], Anke Hoffmann[1], Emilio Ciusani[5], Laura Canafoglia[5], Magdalena Götz[6,7,8] (ID), Rossella Di Giaimo[1,3,*] (ID) & Silvia Cappello[1,**] (ID)

## Abstract

Progressive myoclonus epilepsy (PME) of Unverricht–Lundborg type (EPM1) is an autosomal recessive neurodegenerative disorder with the highest incidence of PME worldwide. Mutations in the gene encoding cystatin B (CSTB) are the primary genetic cause of EPM1. Here, we investigate the role of CSTB during neurogenesis *in vivo* in the developing mouse brain and *in vitro* in human cerebral organoids (hCOs) derived from EPM1 patients. We find that CSTB (but not one of its pathological variants) is secreted into the mouse cerebral spinal fluid and the conditioned media from hCOs. In embryonic mouse brain, we find that functional CSTB influences progenitors' proliferation and modulates neuronal distribution by attracting interneurons to the site of secretion via cell-non-autonomous mechanisms. Similarly, in patient-derived hCOs, low levels of functional CSTB result in an alteration of progenitor's proliferation, premature differentiation, and changes in interneurons migration. Secretion and extracellular matrix organization are the biological processes particularly affected as suggested by a proteomic analysis in patients' hCOs. Overall, our study sheds new light on the cellular mechanisms underlying the development of EPM1.

**Keywords** cystatin B; EPM1; interneuron migration; neurogenesis; secretion
**Subject Categories** Development; Neuroscience; Stem Cells & Regenerative Medicine

## Introduction

EPM1 or Unverricht–Lundborg disease (OMIM #254800) is an autosomal recessive disorder with the onset of symptoms starting between 6 and 13 years of age (Kälviäinen *et al*, 2008). Although EPM1 is a rare disease (most common in Finland with an incidence of 1:20,000; Kälviäinen *et al*, 2008), it is the most common form of progressive myoclonus epilepsy (PME). PME is a group of etiologically and clinically heterogeneous inherited disorders sharing a combination of myoclonus, epilepsy, and progressive neurological deterioration (Berkovic *et al*, 1986). The most common mutation described in EPM1 patients is the expansion of an unstable dodecamer in the *cystatin B* (*CSTB*) promoter which is found in homozygosis (90% of the EPM1 patients) or in heterozygosis together with an allele carrying a point mutation (Lafrenière *et al*, 1997; Lalioti *et al*, 1997; Virtaneva *et al*, 1997; Canafoglia *et al*, 2012). The first two patients with a homozygous R68X mutation in the *CSTB* gene present severe phenotypes with microcephaly and developmental delay starting from 3 months of age in one case (Mancini *et al*, 2016).

CSTB is a widely distributed protein found in most cell types and tissues where it inhibits cysteine proteases, most frequently cathepsin B. Knockout (KO) of *Cstb* in mice generates a neurological disorder with some of the human EPM1 symptoms (Pennacchio *et al*, 1996, 1998). Moreover, neuronal atrophy in the cerebellum (with the Purkinje cells particularly affected), in the cortex, and in the hippocampus is observed both in Cstb-deficient mice and in EPM1 patients (Eldridge *et al*, 1983; Pennacchio *et al*, 1998; Koskenkorva *et al*, 2009). The interpretation of EPM1 patients' histopathological data is complicated by the toxic effect of antiepileptic drugs and the

1 Max Planck Institute of Psychiatry, Munich, Germany
2 International Max Planck Research School for Translational Psychiatry (IMPRS-TP), Munich, Germany
3 Department of Biology, University Federico II, Naples, Italy
4 Department of Environmental, Biological and Pharmaceutical Sciences and Technologies, University of Campania "Luigi Vanvitelli", Caserta, Italy
5 Fondazione IRCCS Istituto Neurologico Carlo Besta, Milan, Italy
6 Physiological Genomics, Biomedical Center (BMC), Ludwig-Maximilians-Universitaet (LMU), Planegg/Martinsried, Germany
7 Helmholtz Center Munich, Biomedical Center (BMC), Institute of Stem Cell Research, Planegg/Martinsried, Germany
8 SyNergy Excellence Cluster, Munich, Germany
  *Corresponding author. Tel: +39 081 2535189; E-mail: digiaimo@unina.it
  **Corresponding author. Tel: +49 8930 622253; E-mail: silvia_cappello@psych.mpg.de
  †The authors contributed equally to this work

degeneration of post-mortem human specimens obtained at late stages of the disease (Koskiniemi *et al*, 1974; Calkins *et al*, 1998; Alakurtti *et al*, 2005). The analyses show non-specific gliosis and neuronal degeneration without intracellular inclusions in many areas of the central nervous system. Besides, a loss of GABAergic neuronal terminals has been described in both the mouse model and human patients (Buzzi *et al*, 2012).

Loss of the antiprotease function of CSTB is the cause of EPM1 (Pennacchio *et al*, 1996), whereas evidence supporting the involvement of cathepsin B is debatable (Di Giaimo *et al*, 2002). Interestingly, the pathological phenotype of *Cstb*-KO mice cannot be efficiently rescued by genetic removal of the target cathepsins (Houseweart *et al*, 2003), suggesting some loss of function other than protease inhibition. On the contrary, the overexpression of another protein of the same cystatin family, cystatin C, rescues degenerating neurons in the *Cstb*-KO mouse model (Kaur *et al*, 2010). Through a yeast two-hybrid system, we have isolated several cytoplasmic proteins that specifically bind Cstb and are involved in cytoskeletal functions and not antiproteolytic function (Di Giaimo *et al*, 2002; Riccio *et al*, 2005; Rispoli *et al*, 2013) Alternative roles, apart from the inhibition of cathepsins, have been described in astrocytes isolated from *Cstb*-KO mice showing increased autophagy (Polajnar *et al*, 2014). A possible function of Cstb in vesicular trafficking and synapse physiology has been proposed by gene expression analyses of granule neurons from *Cstb*-KO mouse (Joensuu *et al*, 2014) and, recently, by studies on the involvement of CSTB in synaptic plasticity (Penna *et al*, 2019).

Taken together, these results suggest that the mouse phenotypes only partially recapitulate the human EPM1 patient phenotype(s).

Here, we study the role of CSTB during brain development by comparing the biological consequences of overexpression of CSTB and a pathological mutant variant in the mouse developing cortex, and in human cerebral organoids (hCOs) (Lancaster & Knoblich, 2015; O'Neill *et al*, 2018; Klaus *et al*, 2019). Moreover, we generated and analyzed patient-specific induced pluripotent stem cells (iPSCs) that we differentiated into hCOs from EPM1 patients with different mutations (Canafoglia *et al*, 2012).

Our results indicate that physiological levels of CSTB are essential for correct cell proliferation and interneurons migration both *in vivo* and *in vitro*. On the contrary, lower levels of functional CSTB in patient-derived hCOs result in a reduction of progenitor cell proliferation, premature differentiation, and reduced recruitment of interneurons. Mechanistically, we found CSTB to be secreted and its downregulation in patient-derived hCOs to hamper the composition of the extracellular environment, as suggested by proteomic analyses. Therefore, we propose a novel role of this protein in the extracellular space.

# Results

## CSTB expression during cortical development in hCOs and mouse developing cortex

To gain insight into the temporal and spatial localization of CSTB, we generated iPSC-derived hCOs and studied its expression across development. *CSTB* transcript is expressed in hCOs in culture, starting on day 16 (d16) until d140 (Fig 1A). The CSTB protein is detected from d40 in hCOs (Fig 1B). It is expressed ubiquitously in both progenitors and neurons as confirmed by gene expression analysis of FACS sorted PAX6$^+$ progenitors and NEUN$^+$ neurons from hCOs at d135 in culture (Figs 1C and EV1A). Single-cell transcriptomic analyses performed in the human fetal cortex confirm the expression of CSTB in progenitor and neurons (Polioudakis *et al*, 2019; Fig EV1B). We also detected Cstb in the cortex and ganglionic eminence of the mouse developing brain from embryonic day 14 (E14) (Fig 1D and E). In accordance with gene expression data (Fig 1C), its expression is ubiquitous in the developing mouse cortex. In particular, it is enriched at the ventricular surface of the cortex, overlapping to stable microtubules (Fig 1E–e‴), in the intermediate zone, and in the cortical plate (Fig 1E).

## CSTB overexpression induces progenitor cell expansion

In order to investigate the role of CSTB during development, we overexpressed GFP-Cstb in hCOs at d35 in culture and in the mouse developing brain (E14) and studied its effect on cell proliferation (Fig 2). The total number of proliferative KI67$^+$GFP$^+$ cells was not changed in hCOs (Fig EV2A) nor mouse cortex (Fig EV2B) but a significant increase in total KI67$^+$ proliferative cells was observed after electroporation of Cstb compared to control in hCOs 5 days post-electroporation (dpe) (Fig 2A and B). Similarly, we observed an increase in total Ki67$^+$ proliferative cells, *in vivo*, in the mouse cortex, at 2 dpe (Fig 2C and D). In order to investigate cell cycle dynamics, we injected intraperitoneally bromodeoxyuridine (BrdU) for 30 min to detect cells in the S-phase. In addition, we performed immunohistochemistry for Ki67 to label the total number of proliferating cells, Phospho-Histone H3 (PH3) to label cells in M-phase, and CyclinD1 (CcnD1), which is specific for G1-phase. Analysis of these different markers revealed a significantly increased number of cells in S-phase (BrdU$^+$Ki67$^+$) upon overexpression of Cstb (Fig 2C and E) but only a trend (not significant) toward an increase in the number of cells in M-phase (PH3$^+$) and G1-phase (CcnD1$^+$) (Figs 2F and EV2C, and EV2D and E).

These results indicate that overexpression of Cstb in the developing mouse cortex results in increased proliferation and affects specifically cells in S-phase.

In order to define which type of progenitor cells is affected upon Cstb overexpression, we analyzed the distinct populations of progenitors, namely radial glia (Pax6$^+$) and intermediate progenitors (Tbr2$^+$) (Figs 2G and EV2F–H). The total number of Ki67$^+$ progenitors was significantly increased, although the specific populations singularly did not show a significant change, despite the clear tendency toward an increase. Interestingly, also the double-positive Pax6$^+$/Tbr2$^+$ cells, in transition between the two stages, show a non-significant increase. This indicates that Cstb is not crucial for cell fate but generally for cell proliferation (Fig 2D).

As the total number of Ki67$^+$ and BrdU$^+$ cycling progenitors is increased (Fig 2E), we analyzed the distribution of cells in S-phase (Fig 2H) and in M-phase (Fig 2I) by subdividing the electroporated cerebral cortex into five equally distributed bins; in both cases, we detected a clear accumulation of cells in Bin3 corresponding to the intermediate zone.

Accordingly, we found both Tbr2$^+$ IPs and Pax6$^+$/Tbr2$^+$ cells accumulating in the same area (Fig 2J and K). On the contrary, the distribution of Pax6$^+$ RGs and GFP$^+$ cells was not altered (Fig EV2I and J).

The ectopic accumulation of cells in Bin3 upon overexpression of Cstb was significantly evident also for PH3$^+$ cells not labeled with

GFP (GFP⁻), suggesting a cell-non-autonomous mechanism (Fig EV2K).

Overall, these results indicate that increased levels of CSTB interfere with physiological cell proliferation during neurogenesis in hCOs and in the developing mouse cortex.

## CSTB is secreted and induces recruitment of migrating interneurons

Recently, we reported that CSTB is detected in synaptic regions and is probably involved in synaptic plasticity since it is locally

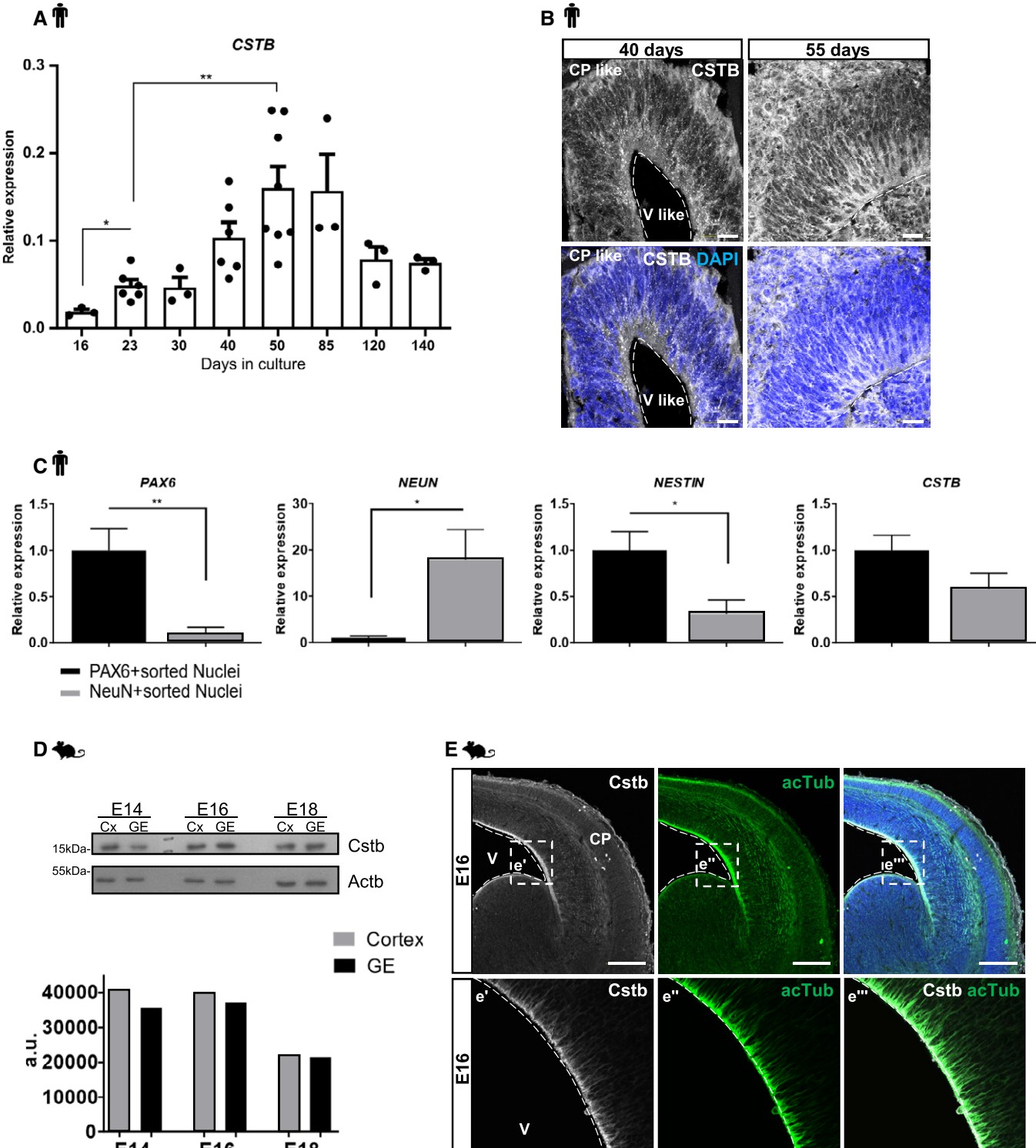

**Figure 1.**

**Figure 1. CSTB expression during cortical development in hCOs and mice.**

A   *CSTB* gene expression analysis in hCOs, starting from day (d) 16 until d140. For every time point, at least 3 different samples were analyzed; each sample was made by a pool of 3–4 hCOs. Data are represented as mean ± SEM. Unpaired *t*-test (*$P < 0.05$, **$P < 0.01$).

B   Micrographs of sections of control hCOs immunostained with CSTB antibody at different time points in culture. The dashed lines represent the apical surface of the cavities/ventricles (V-like). The cortical plate-like side is indicated (CP-like).

C   Real-time qPCR results of *PAX6*, *NEUN*, *NESTIN*, and *CSTB* gene expression levels in PAX6- and NEUN- FACS-sorted nuclei from f-CTRL d135 hCOs. Data are represented as mean ± SEM. Statistical significance was based on Student's *t*-test (*$P < 0.05$, **$P < 0.01$). For PAX6, 10 samples were analyzed as biological replicates: 5 for Pax6[+] sorted nuclei and 5 for Neun[+] sorted nuclei; for NEUN, NESTIN, and CSTB, 9 samples were analyzed as biological replicates: 5 for Pax6[+] sorted nuclei and 4 for Neun[+] sorted nuclei.

D   Western blot analysis and relative quantifications (arbitrary units, a.u.) on protein extracts from cortical (Cx) and ganglionic eminence (GE) tissues for Cstb using Actb as a loading control. Cx and GE tissues from mouse embryos at different time points of development as indicated in the figure.

E   Micrograph of coronal sections of E16 mouse cerebral cortices immunostained for Cstb and acetylated tubulin (acTub) as marker for stabilized microtubules. Ventricle (V) and cortical plate (CP) are indicated. The dashed lines represent the apical surface of the ventricles. Nuclei (blue) are stained with DAPI.

Data information: Scale bars: 50 μm in (B); 100 μm in (E); and 20 μm in (e′, e″, and e‴). Exact *P*-values in Appendix Table S1.
Source data are available online for this figure.

synthesized and secreted from synaptosomes (Penna *et al*, 2019). Therefore, we hypothesized that Cstb is secreted, and thus, we analyzed the CSF of E14 mouse embryos (Fig 3A), and the conditioned medium (CM) of E14 mouse cortical cells and hCOs at different time points (Fig 3B and C) in order to verify its secretion. The analysis of CM revealed the presence of CSTB, but not GAPDH, which was only detectable in protein extract (Fig 3C). Moreover, we overexpressed GFP-Cstb in mouse primary cortical culture demonstrating that also GFP-Cstb is secreted in their CM, while GFP was not detected in CM from cells transfected with the control plasmid (Fig EV3A and B). We previously identified some of the Cstb interactors by using a yeast two-hybrid system (Di Giaimo *et al*, 2002). Among the Cstb partners, we identified kinesin 3 (Kif1a), a molecular motor, and we thus hypothesized a microtubule-dependent transport and/or secretion of Cstb. To verify this hypothesis, we treated E14 mouse primary cells with 100 ng/ml nocodazole, a drug that has been shown to interferes with the polymerization of microtubules (Yang *et al*, 2010). Additionally, we knocked down the expression of Kif1a and Kif5a by transfecting shRNAs targeting both transcripts (Tsai *et al*, 2010a,b). Our results show that the secretion of Cstb is inhibited by nocodazole treatment and by Kif1a downregulation (Fig EV3C and D). Conversely, downregulation of Kif5a has

no effect suggesting a primary role of Kif1a in Cstb secretion (Fig EV3C and D).

Loss of interneuronal GABA synaptic terminals has been observed in both the *Cstb*-KO mouse model and in EPM1 patients (Buzzi *et al*, 2012). To investigate a possible mechanism of Cstb in interneurons, we first speculated that a pathologically low amount of Cstb could interfere with their tangential migration from the ganglionic eminence to the dorsal cerebral cortex (Silva *et al*, 2019). This could explain the imbalance of excitatory/inhibitory neurons, often responsible for seizures. We then investigated if the overexpression of Cstb exclusively in the dorsal forebrain could have a positive effect on the recruitment or distribution of ventrally derived interneurons. To follow the migration of interneurons, we took advantage of two transgenic reporter mouse lines: the GAD65-GFP transgenic mouse line (López-Bendito *et al*, 2004), and GAD67-GFP knock-in mouse line (Tamamaki *et al*, 2003). At E14, we co-electroporated dorsal apical progenitors of the mouse cortex with a plasmid expressing Cstb fused to the HA and a plasmid expressing mCherry, to visualize transfected cells. Three days post-electroporation, we analyzed the number and distribution of interneurons in the cerebral cortex (Fig 3D–G). The total number of GAD65-GFP[+] or GAD67-GFP[+] interneurons did not differ between dorsal areas

**Figure 2. CSTB overexpression induces progenitor cell expansion in hCOs and in the developing mouse cortex.**

A   Micrograph sections of d40 hCOs electroporated with GFP-empty vector control or GFP-Cstb and analyzed 5 dpe. Sections were then immunostained for GFP and KI67. The dashed lines represent the apical surface of the cavities/ventricles (V-like). The cortical plate-like side is indicated (CP-like).

B   Quantification of the total number of proliferating KI67[+] cells/area (μm²) of V-like structures transfected with GFP-empty vector control or GFP-Cstb in (A).

C   Micrograph of coronal sections of E16 mouse cerebral cortices electroporated at E14 with GFP-empty vector or GFP-Cstb, analyzed 2 dpe, and immunostained with GFP, Ki67, and BrdU 30 min after BrdU injection. Ventricle (V) and cortical plate (CP) are indicated. The dashed lines represent the apical surface of the ventricles.

D, E   Quantification of the total number of proliferating Ki67[+] cells/area (μm²) (D) and Ki67[+]BrdU[+] cells/area (μm²) (E) of ventricles transfected with GFP-empty vector or GFP-Cstb in (C).

F, G   Micrograph of coronal sections of E16 mouse cerebral cortices electroporated at E14 with GFP-empty vector or GFP-Cstb, analyzed 2 dpe, and immunostained with GFP and Phospho-Histone H3 (PH3) (F) and with GFP, Pax6, and Tbr2 (G). Ventricle (V) and cortical plate (CP) are indicated. The dashed lines represent the apical surface of the ventricles.

H, I   Distribution of proliferating Ki67[+]/BrdU[+] cells (H) and mitotic progenitor PH3[+] cells (I) in the mouse cortex. The cortex was subdivided into 5 equal bins—Bin1 corresponded to the apical side and Bin5 to the pial side of the cortex and PH3[+] cells or Ki67[+]/BrdU[+] cells calculated in each Bin.

J, K   Distribution of Tbr2[+] intermediate progenitors (J) and Pax6[+]Tbr2[+] cells (K) in the mouse cortex. The cortex was subdivided into 5 equal bins—Bin1 corresponded to the apical side and Bin5 to pial side of the cortex and Tbr2[+] cells or Pax6[+]Tbr2[+] cells calculated in each Bin.

Data information: Scale bars: 50 μm in (A); and 100 μm in (C, F, and G). Data are represented as mean ± SEM. Statistical significance was based on Mann–Whitney test (*$P < 0.05$, **$P < 0.01$). Every dot in the plots refers to independent analyzed ventricles per hCO from at least 3 different hCOs generated in at least 2 independent batches, or independent analyzed mouse brains. Exact *P*-values in Appendix Table S1.
Source data are available online for this figure.

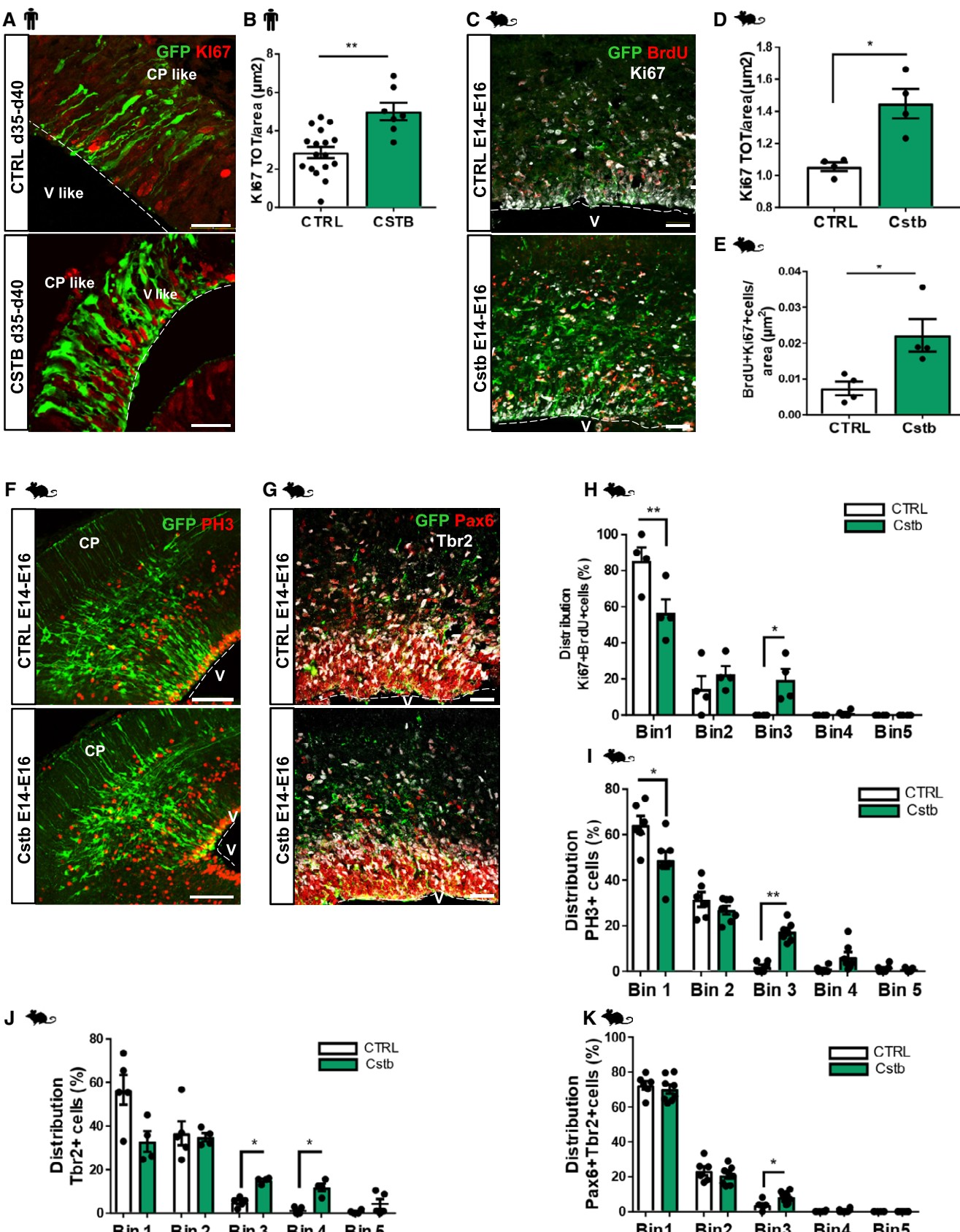

Figure 2.

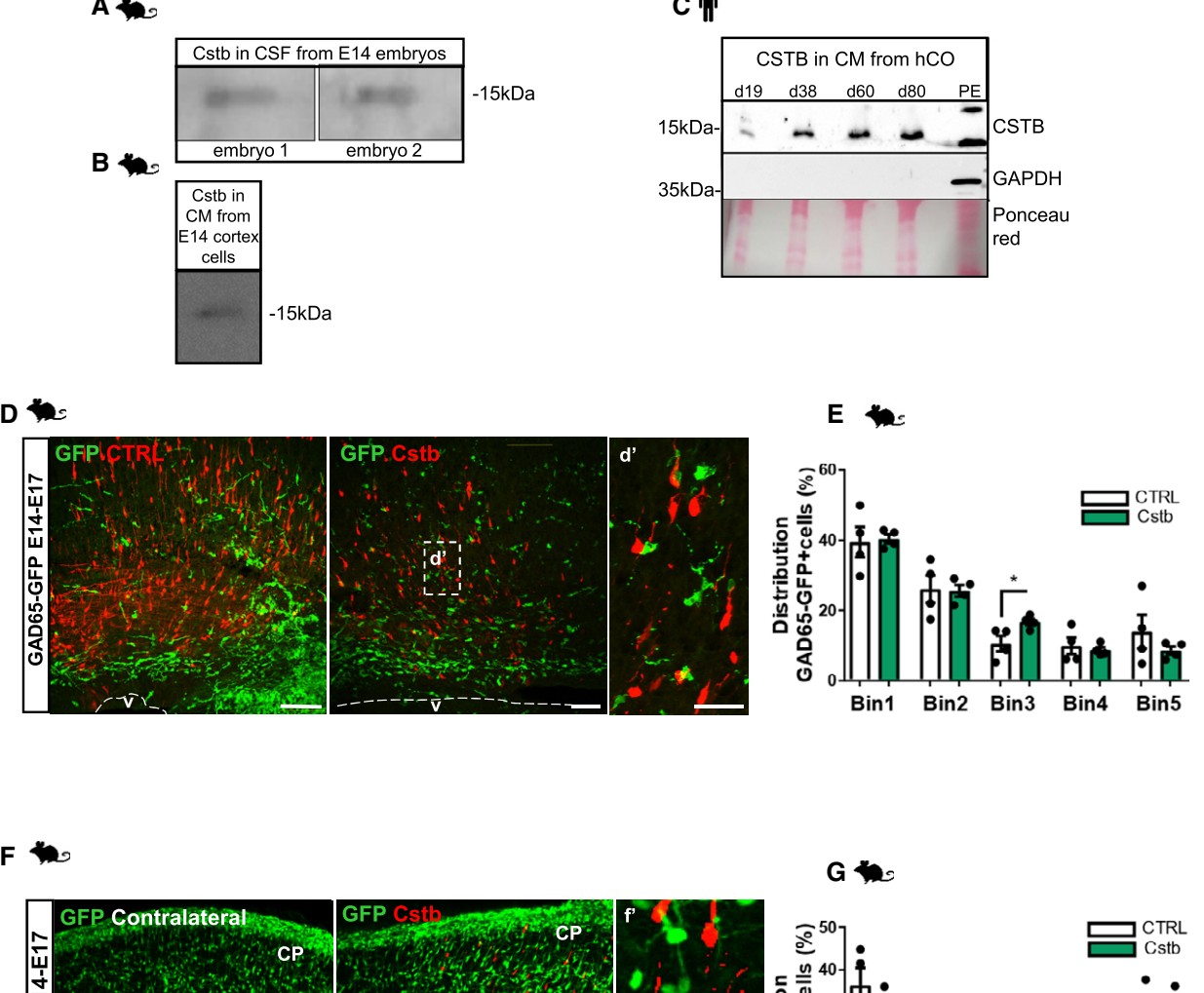

**Figure 3. CSTB is secreted and induces recruitment of migrating interneurons.**

A–C   Western blot analysis for Cstb in the CSF from 2 different E14 mouse embryos (A), in the CM from E14 cortical cells in culture for 4 days (B), in the CM from hCOs after 4 days in culture (C). The CM was collected from hCOs at different time points in culture (d19, d38, d60, d80). Protein extract (PE) from 60 days hCOs was used as a positive control. GAPDH immunostaining was used as a negative control on hCOs CM. Ponceau Red staining is shown to indicate the presence of proteins at comparable levels in the CM samples.

D   Micrograph of coronal sections of E17 mouse cerebral cortices electroporated at E14, co-electroporated with mCherry-expressing vector and HA-empty vector or HA-Cstb, and analyzed 3 dpe. Immunostaining with RFP to identify electroporated cells and GFP to identify migrating interneurons in the GAD65-GFP transgenic mouse line. Ventricle (V) and cortical plate (CP) are indicated. The dashed lines represent the apical surface of the ventricles.

E   Distribution of GFP+ interneurons in the 5 equal bins of the mouse developing cortex—Bin1 corresponded to the apical side and Bin5 to the pial side. Ventricle (V) is indicated. The dashed lines represent the apical surface of the ventricles.

F   Micrograph of coronal sections of E17 mouse cerebral cortices electroporated at E14, co-electroporated with mCherry- and HA-Cstb-expressing vectors, and analyzed 3 dpe. Immunostaining with RFP to identify electroporated cells and GFP to identify migrating interneurons in the GAD67-GFP transgenic mouse line. Ventricle (V) and cortical plate (CP) are indicated. The dashed lines represent the apical surface of the ventricles.

G   Distribution of GFP+ interneurons in the 5 equal bins of the mouse developing cortex comparing electroporated sides with the contralateral ones in the same section to avoid different distribution of interneurons rostro-caudally.

Data information: Scale bars: 100 μm in (D and F) and 20 μm in (d′ and f′). Data are represented as mean ± SEM. Statistical significance was based on Mann–Whitney test (*P < 0.05). Every dot in the plots refers to independent analyzed mouse brains. Exact P-values in Appendix Table S1.
Source data are available online for this figure.

electroporated with the HA-Cstb construct or HA only (Fig EV3E and F). However, in both transgenic lines, upon overexpression of Cstb, we detected a redistribution of the interneurons, resulting in a significant accumulation in the electroporated area (Fig 3D–G, Bin3; and Fig EV3G). In particular, we observed a clear enrichment of (GAD65 and GAD67) GFP$^+$ interneurons in close proximity to Cstb-expressing cells (Figs 3d′–f′ and EV3G), suggesting the involvement of Cstb in the physiological recruitment of interneurons during development.

### Downregulation of *Cstb* results in decreased number and distribution of progenitors

The vast majority of EPM1 patients are either homozygous for expansion (> 30) of the dodecamer repeat in the promoter of *CSTB* or are compound heterozygotes for the expansion of the dodecamer repeat and have a point mutation in the second allele. These mutations cause a pathological reduction of the expression of CSTB (Joensuu *et al*, 2008). To mimic the pathological condition, we reduced the expression of the *Cstb* gene *in vivo* by electroporation of a plasmid expressing a specific microRNA (miRNA) targeting *Cstb* in the mouse developing cortex (Fig 4A). The plasmid was previously validated by immunostaining on primary E14 cortex cells and by qPCR quantification of the transcript (40% decrease in gene expression; Fig EV3H). *Cstb* downregulated (KD) cells are differentially distributed at 3 dpe compared to cells expressing control miRneg (Fig 4A and B). In particular, they accumulate in Bin1 and Bin2, areas of the cortex close to the ventricle, suggesting either a delay in differentiation or a defective migration. While overexpression of Cstb induces increased proliferation in the developing cortex, low levels of Cstb protein had the opposite effect (Fig 4C and D), excluding the delay in differentiation. Similar to what we observed with overexpression, the main effect was not in GFP$^+$ cells, suggesting again a cell-non-autonomous role of Cstb in the developing cortex (Fig EV3I).

Taken together, our results confirm that the amount of Cstb protein in the developing mouse cortex is influencing physiological cell proliferation and neuronal positioning.

### Overexpression of a pathological mutant results in reduced cell proliferation and recruitment of interneurons

EPM1 patients can carry other types of mutations in the *CSTB* gene, including coding regions. For instance, Mancini *et al* (2016) reported one of the patients with a homozygous for nonsense mutation in *CSTB*. The patients, homozygous for a c.202C>T mutation leading to a truncated form of the CSTB protein, named R68X, have severe developmental delay, dyskinesia and microcephaly starting at 3 months of age (Mancini *et al*, 2016). To evaluate the effect of this pathological form of CSTB, we overexpressed R68X and compared the effects with the full-length CSTB. As overexpression and downregulation of Cstb suggested a major role of Cstb in the extracellular space, we first sought to evaluate whether the pathological variant R68X was secreted. Four days post-transfection (dpt), overexpressed R68X in E14 primary cells (Fig EV3J) was not found in the CM by Western blot (WB) (Fig EV3J), while the endogenous Cstb was always detected (Fig EV3A). Notably, the antibody for Cstb recognizes the truncated Cstb form in the total protein extracts

(Fig EV3K) but not in the CM (Fig EV3J). In hCOs, R68X mutant induced a significant decrease in total KI67$^+$ proliferating cells in the electroporated area at 5 dpe, compared to the wt-Cstb and to the control GFP (Fig 4E and F). This result indicates that the truncated form of the protein is sufficient to abolish the effect on cell proliferation obtained with the full-length protein. Moreover, the R68X mutant results in decreased proliferation (even compared to control) (Fig 4E and F), similarly to the downregulation (Fig 4C and D), suggesting a dominant effect. Indeed, R68X has also a direct effect, as indicated by the significant decrease in the KI67$^+$GFP$^+$ cell population (Fig 4G). On the contrary, we did not observe cell-autonomous effects upon overexpression of the wt-Cstb (Fig EV2A) or upon its downregulation in the mouse developing cortex (Fig EV3I). However, R68X mutation results in accumulation of Doublecortin$^+$ (DCX$^+$) newborn neurons at the ventricle of mouse embryonic cortex, not detected upon interference with wt-*Cstb* expression levels (Fig EV3L and M). R68X overexpression resulted in failure of the correct positioning of the GFP$^+$ electroporated cells that accumulate in Bin1 at the expense of Bin2 and Bin3 (Fig 4H and I) and PH3$^+$ mitotic cells (Fig 4J). However, the number of PH3$^+$ cells was not affected (Fig 4K). Finally, also the total number of Tbr2$^+$ IPs was decreased upon R68X overexpression compared to the full-length Cstb (Fig 4L and M).

As previously suggested, forced expression of Cstb induces a redistribution of interneurons. Therefore, we tested whether the truncated R68X retains this function on interneurons. On the contrary, R68X overexpression interfered with the migration of interneurons (Fig 4N and O), again strongly suggesting a dominant effect of this pathological form of CSTB. Overall, these results indicate the opposite effects of wt-CSTB and the pathological R68X variant on cell proliferation and migration in humans and mice.

### Proliferation is reduced in EPM1-derived cerebral organoids in a cell-non-autonomous manner

In order to gain more insights into the pathological neurogenic phenotype of EPM1 patients, we generated hCOs from patients' cells. We collected blood samples from a control individual (b-CTRL) and two different EPM1 patients; the first bearing a homozygous amplification in the *CSTB* promoter (UL1) and the second being compound heterozygote for the dodecamer repeat and a point mutation leading to skipping of exon 2 (UL4) (Pennacchio *et al*, 1996; Joensuu *et al*, 2008). We reprogrammed human peripheral blood mononuclear cells (PBMCs) to obtain iPSCs that were then used to generate hCOs (Fig EV4A). By real-time qPCR and WB, we showed that the expression levels of *CSTB* mRNA and protein were strongly reduced in the UL1 and UL4 iPSCs compared to two different control iPSC lines (Fig EV4B and C), one derived from PBMCs (b-CTRL) and one from fibroblasts (f-CTRL) (Klaus *et al*, 2019).

The parallel generation of cerebral organoids from two controls (b-CTRL and f-CTRL) and two EPM1 patients (UL1 and UL4) showed that UL1 and UL4 organoids were significantly smaller than controls (Fig 5A). As suggested by our previous results, we first investigated cell proliferation and analyzed the number of PH3$^+$ and KI67$^+$ cells in the proliferative regions of the hCOs derived from patients and controls (Fig 5B–G). Similarly, to the *Cstb* KD, we detected a significant decrease in proliferating KI67$^+$ and PH3$^+$ cells in UL1 and UL4 hCOs. Interestingly, the decrease in proliferation

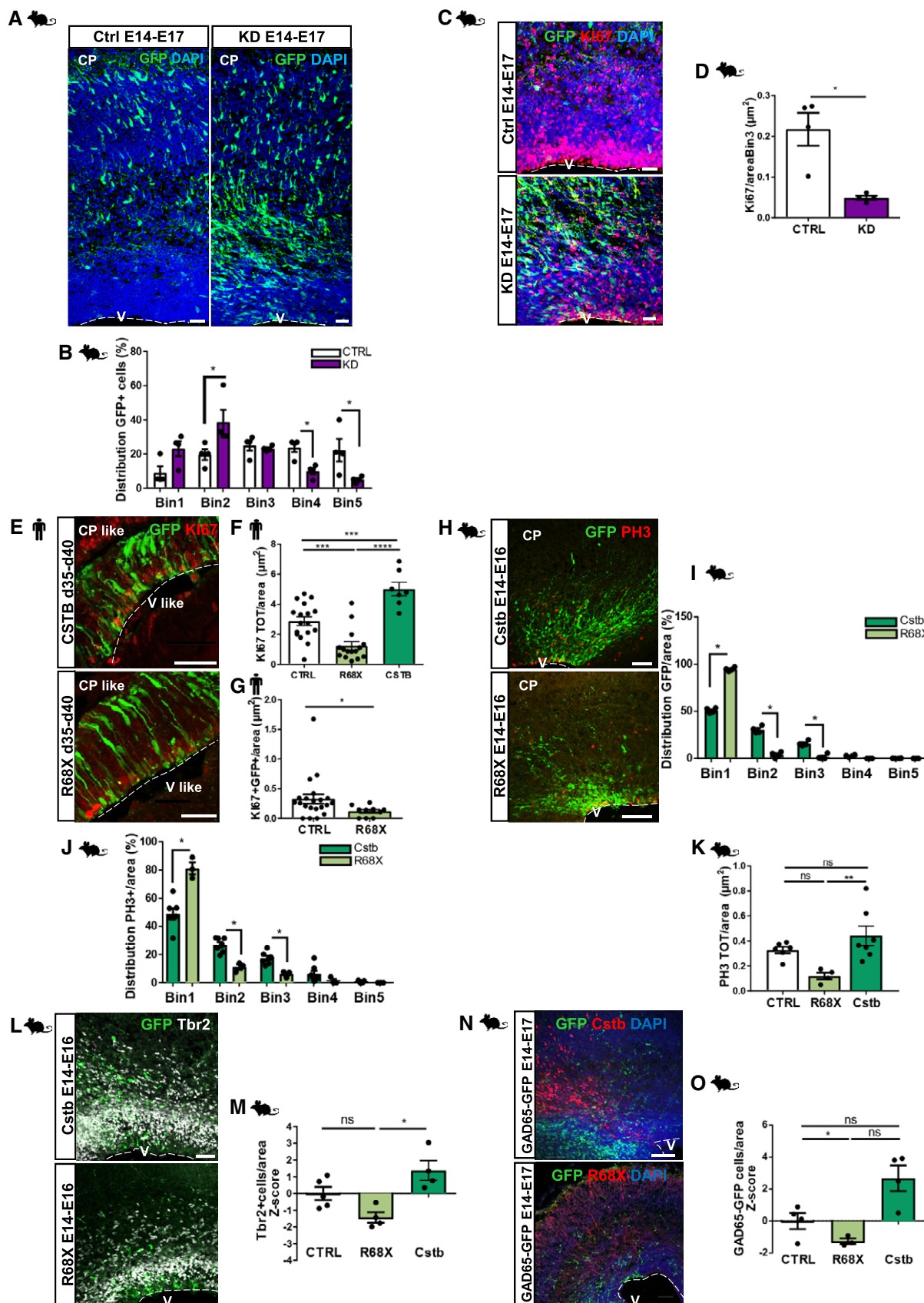

Figure 4.

◄

**Figure 4. Downregulation of *Cstb* and R68X overexpression results in a decreased number of progenitors and migrating interneurons in the developing mouse cortex.**

A  Micrograph of coronal sections of E17 mouse cerebral cortices electroporated at E14 with GFP-miR-neg vector (CTRL) and GFP-miRNA-*Cstb* (KD), analyzed 3 dpe, and immunostained with GFP. Ventricle (V) and cortical plate (CP) are indicated. The dashed lines represent the apical surface of the ventricles.

B  Distribution of electroporated GFP$^+$ cells in the mouse cortex. The cortex was subdivided into 5 equal bins—Bin1 corresponded to the apical side and Bin5 to the pial side of the cortex and GFP$^+$ cells calculated in each Bin.

C  Micrograph of coronal sections of E17 mouse cerebral cortices electroporated at E14 with GFP-empty vector control and GFP-miRNA-*Cstb*, analyzed 3 dpe, and immunostained with GFP and Ki67. Ventricle (V) is indicated. The dashed lines represent the apical surface of the ventricles.

D  Quantification of the total number of proliferating Ki67$^+$ cells/area (μm$^2$) of ventricular structures transfected with GFP-miR-neg vector or GFP-miRNA-*Cstb* in (C).

E  Micrograph sections of d40 hCOs electroporated with GFP-Cstb-expressing vector or GFP-R68X mutant and analyzed 5 dpe. Sections were immunostained for GFP and Ki67. The dashed lines represent the apical surface of the cavities/ventricles (V-like). The cortical plate-like side is indicated (CP-like).

F, G  Quantification of the total number of proliferating KI67$^+$ cells/area (μm$^2$) (F) and Ki67$^+$GFP$^+$ cells/area (μm$^2$) (G) of ventricle-like structures transfected with GFP-empty vector control, GFP-Cstb or GFP-R68X mutant in (E).

H  Micrograph of coronal sections of E16 mouse cerebral cortices electroporated at E14 with GFP-empty vector control, GFP-*Cstb*, or GFP-R68X, analyzed 2 dpe, and immunostained with GFP and Phospho-Histone H3 (PH3). Ventricle (V) and cortical plate (CP) are indicated. The dashed lines represent the apical surface of the ventricles.

I, J  Distribution of electroporated GFP$^+$ cells (J) and mitotic progenitor PH3$^+$ cells (K) in the mouse cortex. The cortex was subdivided into 5 equal bins—Bin1 corresponded to the apical side and Bin5 to the pial side of the cortex and GFP$^+$ cells or PH3$^+$ cells calculated in each Bin.

K  Quantification of the total number of proliferating PH3$^+$ cells/area (μm$^2$) of ventricular structures transfected with GFP-empty vector control, GFP-*Cstb*, or GFP-R68X mutant in (H).

L  Micrograph of coronal sections of E16 mouse cerebral cortices electroporated at E14 with GFP-empty vector control, GFP-*Cstb*, or GFP-R68X mutant, analyzed 2 dpe, and immunostained with GFP and Tbr2. Ventricle (V) is indicated. The dashed lines represent the apical surface of the ventricles.

M  Quantification of the total number of Tbr2$^+$ cells/cortical area (μm$^2$) in (L). Data shown as Z-scores relative to the mean of GFP control vector.

N  Micrograph of coronal sections of E17 mouse cerebral cortices electroporated at E14, co-electroporated with mCherry-expressing vector and HA-C*stb* or R68X, and analyzed 3 dpe. Immunostaining with RFP to identify electroporated cells and GFP to identify migrating interneurons in the GAD65-GFP transgenic mouse line. Ventricle (V) is indicated. The dashed lines represent the apical surface of the ventricles.

O  Quantification of the total number of GAD65-GFP interneurons/cortical area (μm$^2$) in (N). Data shown as Z-scores relative to the mean of GFP control vector.

Data information: Nuclei (blue) are stained with DAPI. Scale bars: 50 μm in (A, C, E, and H); 100 μm in (L and N). Data are represented as mean ± SEM. Statistical significance was based on Mann–Whitney test (*$P < 0.05$, **$P < 0.01$, ***$P < 0.001$, ****$P < 0.0001$). Every dot in the plots refers to independent analyzed ventricles per hCO from at least 3 different hCOs generated in at least 2 independent batches, or independent analyzed mouse brains. Exact P-values in Appendix Table S1. Source data are available online for this figure.

only became significant at d60 for UL4 (Fig 5E), suggesting different kinetics in different pathological variants.

Additionally, cell proliferation was assessed in control and patients' cells derived from dissociated hCOs, at d30 and d60 in culture, and cultured cells for 3 days (Fig EV4D–F). Supporting the previous findings, in both UL1 and UL4 patients' cells, we detected a decrease in KI67$^+$ cells at earlier (d30) and later (d60) time points (Fig EV4E and F).

In order to verify the possible cell-non-autonomous role of CSTB in hCOs, CM from hCO cultures was swapped between CTRL and patients' hCOs every day (Fig 5H). After 36 days in culture, proliferation was analyzed by FACS using Ki67 (Fig EV4G) and PH3 (Fig EV4H and I). Strikingly, CTRL hCOs show a significant decrease in proliferating Ki67$^+$ (Fig 5F) and PH3$^+$ cells (Fig EV4I) when exposed to CM from patients' hCOs. Conversely, patients' hCOs cultured in CTRL CM showed a significant increase in Ki67$^+$ cells (Fig 5F). Once exposed to CTRL CM, the total number of Ki67$^+$ cells was rescued, as it was not significantly different from CTRL hCOs.

These results clearly indicate that secreted CSTB (and/or molecules secreted depending on CSTB levels in the cell) can affect the physiological proliferation in a cell-non-autonomous manner.

## EPM1-derived cerebral organoids exhibit premature differentiation

To investigate if the decrease in the proliferating cells could result in a change of neuronal output, we quantified the number of DCX$^+$ young neurons at d30 and d60 hCOs using immunostaining (IHC)

and FACS analysis (Fig 6). DCX immunopositive cells form a wider band in the presumptive cortical plate region in both UL1 and UL4 hCOs compared to controls at both developmental time points (Fig 6A–D). This increase in DCX$^+$ neurons was confirmed by FACS analysis of d60 hCOs (Figs 6E and EV5A). Moreover, the number of Tubulin Beta 3 (TUBB3$^+$) neurons derived from dissociated organoids at d30 and d60, was increased (Fig EV5B–D).

Our *in vivo* data support the hypothesis that CSTB and its pathological variant are involved in the regulation of the recruitment and correct positioning of interneurons. Therefore, to validate this function in a human model system, we took advantage of a recently developed protocol that combines patterned ventral and dorsal organoids to specifically reproduce *in vitro* the dorso-ventral forebrain axis and the tangential migration of ventral interneurons occurring during development (Bagley *et al*, 2017). We generated patterned dorsal organoids with either f-CTRL or UL1 cells and patterned ventral organoids with f-CTRL cells, previously stably transfected with GFP, to follow the trajectory of the GFP$^+$ control interneurons toward the dorsal organoids. We then fused the organoids (Bagley *et al*, 2017) to obtained fused organoids composed of ventrally derived control neurons and dorsally derived control or patients' cells (Fig 6F). Dorsal organoids were then analyzed at d30 and GFP$^+$ ventral cells were quantified starting from a 100-μm distance from the dorso-ventral boundary (Fig 6G). The number of ventral GFP$^+$ cells that migrated deeper in the dorsal area was significantly decreased in the dorsally patterned UL1 organoids compared to the dorsally patterned f-CTRL organoids (Fig 6H), suggesting a role of Cstb in the process of recruitment of interneurons also in humans.

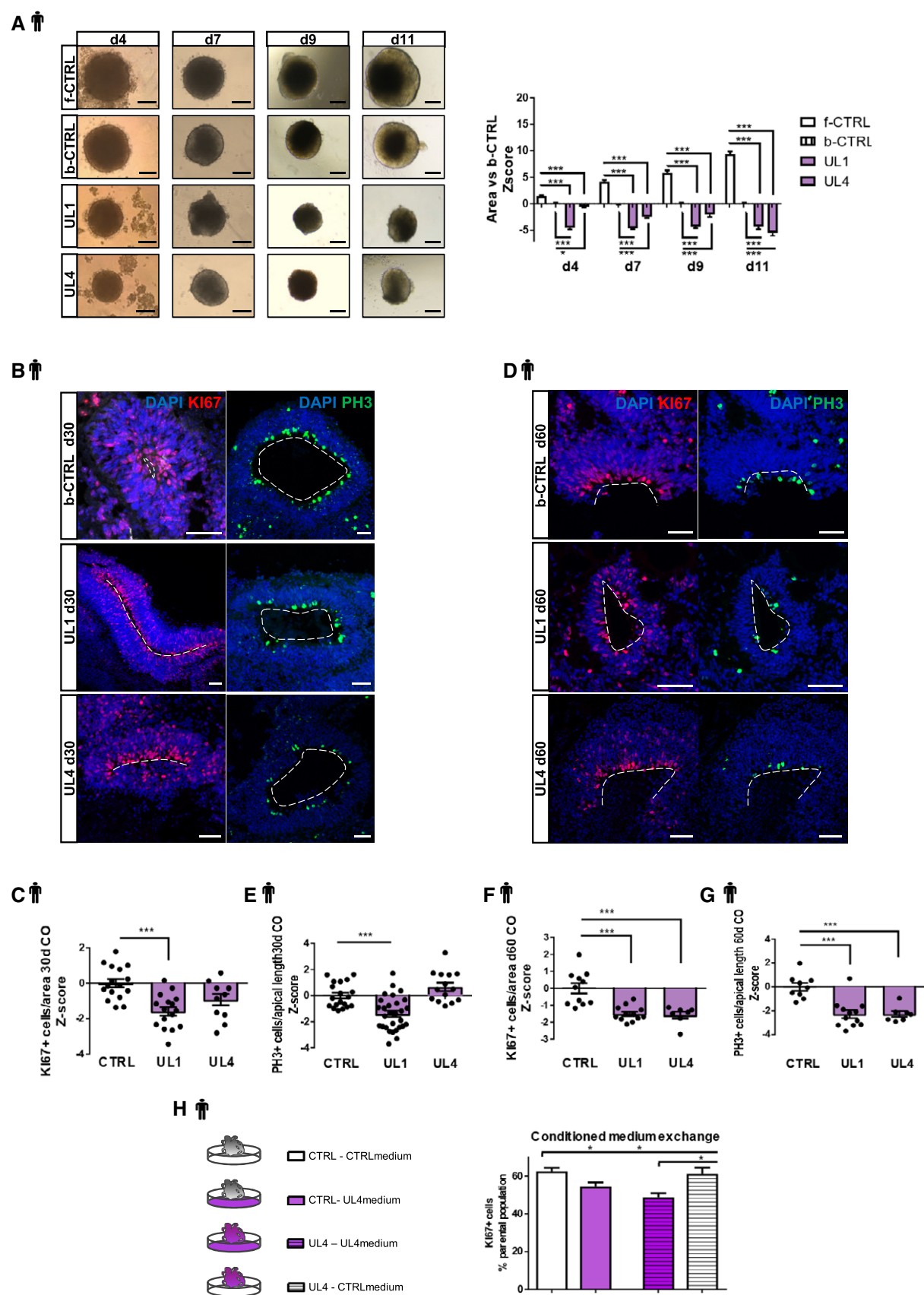

Figure 5.

◀

**Figure 5.  Proliferation is reduced in EPM1-derived cerebral organoids in a cell-non-autonomous manner.**

A    The left panel shows representative images of hCOs from the different iPSC lines at the analyzed time points. The right panel shows the difference of the areas of f-CTRL, b-CTRL, UL1, and UL4 hCOs at different time points in culture. Data shown as Z-scores relative to the mean of b-CTRL.

B    Micrograph sections of d30 b-CTRL, UL1, and UL4 hCOs immunostained for proliferation markers KI67 and PH3. The dashed lines represent the apical surface of the cavities/ventricles.

C, D  Quantification of the total number of proliferating Ki67$^+$ cells/area ($\mu m^2$) of ventricle-like structures and of the number of PH3$^+$ cells facing the ventricle/apical surface length ($\mu m$). Data shown as Z-scores relative to the mean of b-CTRL.

E    Micrograph sections of d60 b-CTRL, UL1, and UL4 hCOs immunostained for proliferation markers KI67 and PH3. The dashed lines represent the apical surface of the cavities/ventricles.

F, G  Quantification of the total number of proliferating KI67$^+$ cells/area of ventricle-like structures and of the number of PH3$^+$ cells facing the ventricle/the apical surface length ($\mu m$). Data shown as Z-scores relative to the mean of CTRL.

H    Scheme of the experiment with conditioned medium exchange between b-CTRL and UL4 hCOs and quantification of the Ki67$^+$ cells by FACS analysis on 36 days old dissociated hCOs. Four samples were analyzed as biological replicates. FACS plots in Fig EV4G.

Data information: Nuclei (blue) are stained with DAPI. Scale bars: 200 $\mu m$ in (A); and 50 $\mu m$ in (B and E). Data are represented as mean ± SEM. Statistical significance was based on Mann–Whitney test (*$P < 0.05$, ***$P < 0.001$). Every dot in the plots refers to independent analyzed ventricles per hCO from at least 3 different hCOs generated in at least 2 independent batches. Exact $P$-values in Appendix Table S1.

Source data are available online for this figure.

Taken together, these results shed new light on the role of CSTB during neurogenesis suggesting that CSTB is involved in correct recruitment and migration of interneurons, an essential developmental process often dysregulated in patients with epilepsy. These data provide a possible mechanism for the loss of GABAergic neuronal terminals described in both the EPM1 mouse model and human patients (Buzzi et al, 2012).

To investigate the molecular mechanism(s) underlying the striking cell-non-autonomous phenotypes, we performed a TMT-based quantitative mass spectrometry analysis of the proteins from control and patient's organoids at 40 days of development in culture. The mass spectrometry proteomics data have been deposited to the ProteomeXchange Consortium via the PRIDE (Perez-Riverol et al, 2019) partner repository with the dataset identifier PXD018021. We found 178 differentially represented proteins (Figs 6I and EV5E; Table EV1). As expected, some of these proteins belong to biological processes that were already described for the EPM1 disease like oxido-reduction stress (Lehtinen et al, 2009; Butinar et al, 2014; Fig 6J). Besides, oligomerization and aggregation of CSTB have been reported as sensitive to redox status of the environment (Ceru et al, 2010; Rispoli et al, 2013). However, we found de-regulated biological processes, namely secretion, cell–cell adhesion, and protein–lipid assembly, that indicate a general effect of CSTB deficiency on the mechanisms affecting secretory pathways and cell–cell contact.

To gain more insight, we focus on cell-type-specific human expression data (single-cell RNA-seq from human brain fetal cortex; Polioudakis et al, 2019; http://solo.bmap.ucla.edu/shiny/webapp/; Fig EV5F). We then clustered the 178 differentially expressed proteins in four categories (Fig EV5F) and focused on the proteins enriched in radial glial cells (RGs) that represent 7% of the total.

Gene Ontology analysis of the 12 proteins that are RG-specific (Table EV2) indicates that EPM1 hCOs show an altered set of proteins that participate to the formation of the extracellular matrix (Fig EV5G). This result strongly suggests that the neural progenitors are altered in the disease, and this can affect cell proliferation and migration during cortical development. Furthermore, five out of the 12 RG-specific proteins (COL11A1, RRAS, GLUD1, IFITM3, and MGST1) are involved in human epileptic disorders prompting the hypothesis that modification of the extracellular environment is a common feature in this type of brain disease.

# Discussion

### Role of CSTB during brain development

*CSTB* is the gene that is most often mutated in reported cases of EPM1 disease, and the severity of EPM1 is inversely correlated with the amount of residual functional CSTB protein. EPM1 has been mostly attributed to neuronal apoptosis triggered by cathepsins' activity, physiologically inhibited by CSTB. However, the precise function of CSTB in the developing and adult brain and the relationship between the deficiency of CSTB and the clinical symptoms still remain unknown. The identification of increased Cstb expression in a rat model of epilepsy (D'Amato et al, 2000) and neuronal cellular atrophy in the cortex of *Cstb*-KO mice indicate that Cstb has a crucial role in the maintenance of normal neuronal architecture. Moreover, other cellular dysfunctions like oxidative damage due to the lack of Cstb may contribute to the pathological phenotype (Shannon et al, 2002; Lehtinen et al, 2009; Žerovnik, 2019). Functional MRI studies performed on brains of individuals with EPM1 show atrophic changes in the cerebral cortex and, marginally, in the cerebellum. These results indicate that deficiency of CSTB mostly affects the cortical areas of the brain (Nigri et al, 2017). Due to the early clinical manifestations of the EPM1 patients (usually at the age of 6 but sometimes even at the age of 3 months), we hypothesized a role of CSTB during the early stages of cortical development.

Interestingly, in hCOs, the low amount of CSTB detected in patient-derived organoids did not interfere with the general neurogenic program and hCOs developed from both of the EPM1-derived iPSC lines studied here. Nevertheless, critical levels of CSTB are essential for the correct expansion of the neural progenitors and prevention of premature differentiation.

Moreover, *in vivo*, both overexpression and downregulation of *Cstb* in the developing cortex influence physiological cell proliferation. Therefore, we demonstrated that CSTB has a function during cortical development, and these data may elucidate the cellular mechanism underlying the reduction in cortical thickness seen in patients with EPM1 and in the *Cstb*-KO mice (Koskenkorva et al, 2009; Danner et al, 2013).

On the contrary, downregulation or overexpression of a pathological variant of CSTB impacts cell proliferation negatively.

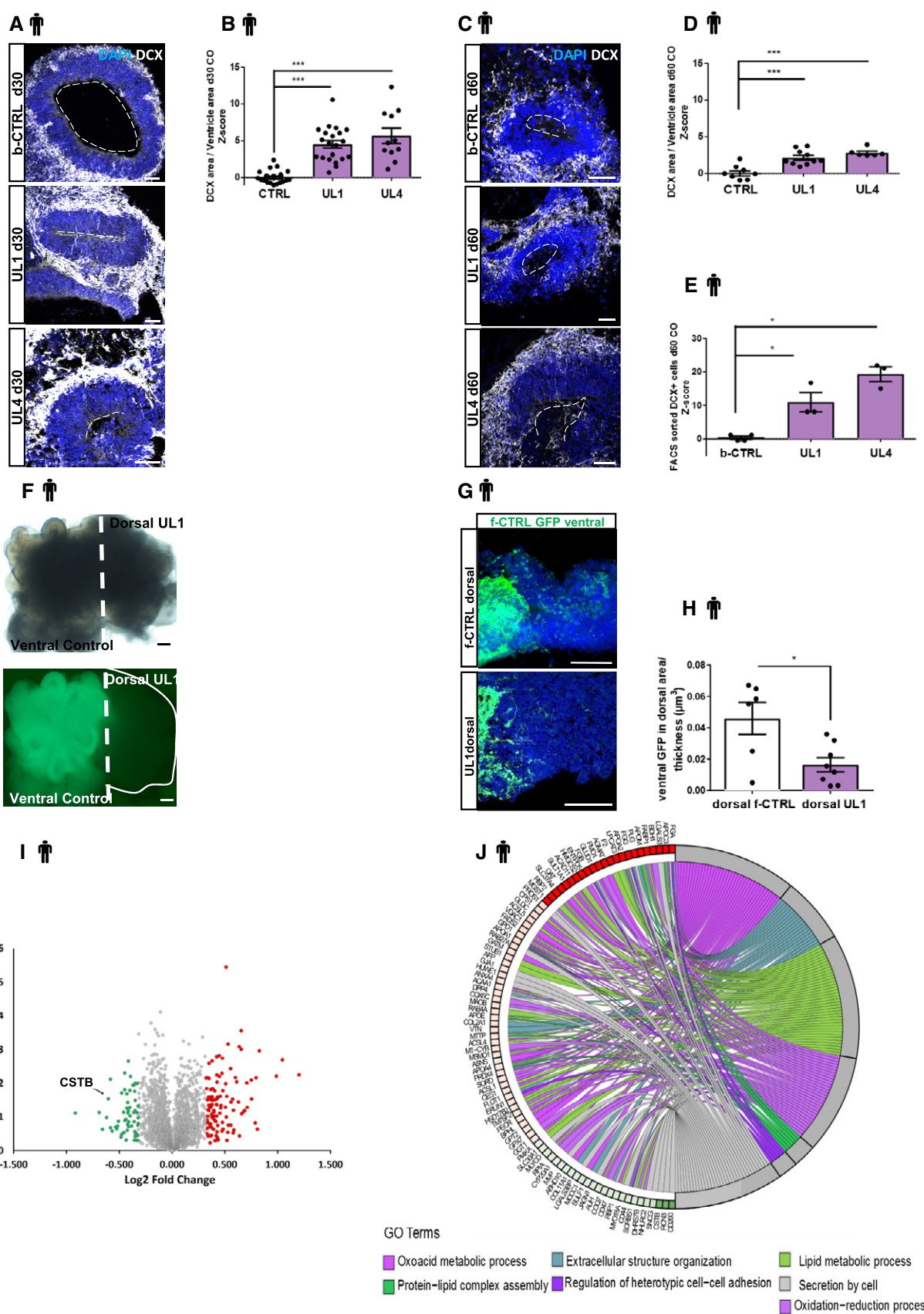

**Figure 6.**

**Figure 6. EPM1-derived cerebral organoids exhibit premature differentiation. Proteomic analysis in patients' hCOs.**

A   Micrograph sections of d30 b-CTRL, UL1, and UL4 hCOs immunostained for the neuronal marker DCX. The dashed lines represent the apical surface of the cavities/ventricles.

B   Quantification of the DCX$^+$ area. The area of DCX$^+$ cells was calculated and related to the total area ($\mu m^2$) of the ventricular structure. Data shown as $Z$-scores relative to the mean of b-CTRL.

C   Micrograph sections of d60 b-CTRL, UL1, and UL4 hCOs immunostained for neuronal marker DCX. The dashed lines represent the apical surface of the cavities/ventricles.

D   Quantification of the DCX$^+$ area. The area ($\mu m^2$) of DCX$^+$ cells was calculated and related to the total area of the ventricular structure. Data shown as $Z$-scores relative to the mean of b-CTRL.

E   Quantification of the number of neuronal DCX$^+$ cells by FACS sorting on 60 days old dissociated hCOs. Data shown as $Z$-scores relative to the mean of b-CTRL. FACS plots in Fig EV5A.

F   Micrograph sections of d30 fused organoid; ventral GFP-f-CTRL and dorsal UL1.

G   Micrographs of sections from d30 fused organoids, showing ventral GFP$^+$ cells in the dorsal area. The migrating GFP$^+$ cells were counted starting from 100 $\mu m$ far from the ventral/dorsal border.

H   Quantification of ventrally derived GFP$^+$ cells in dorsal/thickness of the counted area ($\mu m^3$). The counted area was always 100 $\mu m$ far from the border between ventral and dorsal area as indicated in Fig EV5D. Every dot represents a different hCOs and different areas were counted for each hCO.

I   Volcano plot representing proteins identified by high-resolution nanoLC MS/MS analysis. Up- and downregulated proteins ($-0.3 \geq \log_2 FC \geq 0.3$) in human cerebral organoids (UL1 vs. b-CTRL) are shown in red and green, respectively.

J   GOChord plot showing relationships between selected representative biological process GO terms and differentially expressed proteins identified by nanoLC MS/MS analysis. The representation shows up (red)- and downregulated (green) proteins differentially expressed in UL1 relative to b-CTRL ($-0.3 \geq \log_2 FC \geq 0.3$).

Data information: Nuclei (blue) are stained with DAPI. Scale bars: 50 $\mu m$ in (A and C); 200 $\mu m$ in (F); and 100 $\mu m$ in (G). Data are represented as mean $\pm$ SEM. Statistical significance was based on Mann–Whitney test (*$P < 0.05$, ***$P < 0.001$). Every dot in the plots refers to independent analyzed ventricles per hCO from at least 3 different hCOs generated in at least 2 independent batches. Exact $P$-values in Appendix Table S1.

Source data are available online for this figure.

Accordingly, patients with low levels of the protein show a reduced brain volume. More strikingly, homozygous patients with non-functional pathological variants of CSTB (R68X) are microcephalic (Mancini *et al*, 2016).

### Cell-non-autonomous role of secreted CSTB: interneurons migration and epilepsy

By demonstrating that Cstb is secreted into the CSF, we now open new avenues for understanding the striking cell-non-autonomous function(s) of CSTB. We show that both cell proliferation and interneuron migration are impaired *in vivo* and in hCOs in cells that have physiological levels of Cstb/CSTB (non-transfected). To demonstrate the role of secreted CSTB, we switched the conditioned medium of control hCOs and patient's organoids. This simple exchange was sufficient to rescue the defect in cell proliferation described in patient-derived hCOs.

The finding that the secretion of CSTB promotes also the tangential migration and recruitment of interneurons of ventral origin in the mouse developing cortex and dorsally patterned hCOs could explain why the pathological decrease in CSTB creates a condition of latent hyperexcitability that favors myoclonus and seizures. Hence, perturbations of interneurons migration, maturation, and integration into cortical circuits can lead to an imbalance of cortical excitation and inhibition and this contributes to the insurgence of neuropsychiatric disorders, including epilepsy and autism spectrum disorders (ASD) (Rubenstein & Merzenich, 2003; Marín, 2012). Accordingly, the proteomic analysis, performed in COs, highlights cell secretion as one of the biological processes altered in patient hCOs compared to controls. Moreover, many proteins involved in the formation and maintenance of the extracellular matrix are altered (e.g., collagens, vitronectin, and surface proteins that associate with tetraspanins). These novel findings provide possible mechanisms for altered neurogenesis and neuronal migration driven by low levels of CSTB.

### Additional players in EPM1 and epilepsies

It is still unclear how different types of progressive myoclonus epilepsies can have synergistic mechanisms since mutations in two other genes, *PRICKLE1* (Prickle planar cell polarity protein 1) and *SCARB2* (Scavenger receptor class B member 2), have been reported in patients with symptoms closely resembling EPM1 (Bassuk *et al*, 2008; Dibbens *et al*, 2009). PRICKLE1 contributes to neuronal morphogenesis, including axonal and dendritic extension during mouse brain development (Liu *et al*, 2013) and localizes adjacent to focal adhesions to promote cell migration (Lim *et al*, 2016). Interestingly, cell adhesion is one of the affected biological processes in our proteomic analysis of patients' hCOs. Furthermore, focal adhesions connect the cytoskeleton to the extracellular matrix (Parsons *et al*, 2010), suggesting that the pathway(s) that are hampered in EPM1 can be in common between different mutated genes. SCARB2 participates in membrane transport and reorganization of the endosomal/lysosomal compartment (Reczek *et al*, 2007). Accordingly, proteomic analyses of CSTB patients' hCOs have altered secretory mechanisms, suggesting other possible converging mechanisms.

It is also interesting to outline that almost half of the proteins that are differentially expressed in EPM1-derived organoids and enriched in radial glial cells are involved in different types of human epilepsies. For instance, Collagen COL11A1 mutated in the Stickler syndrome, caused by mutations in COL2A1 or COL11A1, is associated with epileptic phenotypes (Savasta *et al*, 2015; Zhou *et al*, 2018); IFITM3 (interferon-induced transmembrane protein 3) has been identified among genes that are target of epilepsy-associated miRNA in animal models and human tissues (Cava *et al*, 2018); heterozygous mutation in GLUD1 gene has been reported in patients affected by hyperinsulinism, commonly associated with epileptic seizures (Aka *et al*, 2016); RRAS (Guo *et al*, 2017) and MGST1 (Shang *et al*, 2008; Ercegovac *et al*, 2015) are associated with epilepsy risk.

Taken together, our proteomic results support the novel concept that remodeling of extracellular matrix and contacts between cells are critical processes for epileptogenesis (Dityatev, 2010).

Overall, our data reveal a role for CSTB in regulating the early stages of human neurogenesis. CSTB levels are particularly critical for the maintenance of an appropriate cellular environment in the developing brain. This includes a role of CSTB as a long-distance signaling molecule crucial for the recruitment of inhibitory neurons, uncovering a possible mechanism through which altered levels of CSTB results in epilepsy.

The possibility to rescue some of the pathological aspects emerged in patients' hCOs by suppressing the signaling triggered by the remodeled extracellular matrix (ECM), might suggest effective therapeutic strategies to antagonize the progression of the pathological phenotype.

# Materials and Methods

## Reprogramming of PBMCs from patients and control human samples

PBMCs were isolated from human blood by using SepMate-50 tubes (StemCell, 15450) according to the manufacturer's instructions. PBMCs were collected from (i) a control female sample, 39 years old (b-CTRL); (ii) an EPM1 female patient sample, 35 years old (UL1); and (iii) an EPM1 female patient sample, 13 years old (UL4). iPSCs were reprogrammed from PBMCs as previously reported (Dowey et al, 2012; Chou et al, 2015) with some modifications.

iPSCs were generated with the approval of the Ethical commission, Ludwig Maximilian University of Munich, Germany (Project 19-635). Informed consent was obtained from all subjects and the experiments conformed to the principle set out in the WMA Declaration of Helsinki and the Department of Health and Human Services Belmont Report.

Day 0: A minimum number of $5 \times 10^6$ PBMCs were cultured in a 25-cm$^2$ flask containing 5 ml Erythroid Expansion Medium (IMDM 49% v/v, Life Tech 21056023; Ham's F-12 49% v/v, Life Tech 11765-054; ITS-X 1% v/v, Life Tech 51500-056; lipid concentrate 1% v/v, Life Tech 11905031; L-Glutamine 1% v/v, Life Tech 25030024; L-Ascorbic acid 0,005% w/v, Sigma A8960; BSA 0.5% w/v, Sigma A9418; 1-thioglycerol 0.0018% v/v, Sigma M6145; human stem cell factor (SCF) 100 ng/ml, PeproTech 300-07; human IL-3 10 ng/ml, PeproTech 200-03; IGF-1 40 ng/ml, PeproTech 100-11; Erythropoietin (EPO) 2 units/ml, KMS (Janssen-Cilag); holo-transferrin 0.1 mg/ml, R&D Systems 2914-HAT-100MG; Dexamethasone 1 μM, Sigma D4902).

Day 1: Cell suspension was transferred into a new 25-cm$^2$ flask. Days 2, 4, and 6: Cells were transferred into a 15-ml conical tube and centrifuged at $300 \times g$ for 5 min. Supernatant was removed, and cells were resuspended in 1 ml fresh Erythroid Expansion Medium and plated in a new 25-cm$^2$ flask containing 5 ml fresh Erythroid Expansion Medium.

Day 8 (day of transfection): Cells were collected in a 15-ml conical tube, centrifuged at $300 \times g$ for 5 min and the cell pellet resuspended in 2 ml fresh Erythroid Expansion Medium.

The cell number was determined using the LUNA-II Automated Cell Counter from Logos Biosystems according to the manufacturer's instructions.

For each nucleofection, $2 \times 10^6$ culture-expanded erythroid cells were transferred into a 15-ml conical tube and centrifuged at $300 \times g$ for 5 min, the supernatant was carefully removed and cells were resuspended in 100 μl CD34$^+$ Cell Nucleofector Solution (Lonza, VAPA-1003). 4 μg MOS (Addgene, 64120), 4 μg MMK (Addgene, 64121), and 2 μg GBX (Addgene, 64123) were added to the cell suspension and mixed well. Cells were electroporated with the reprogramming vectors using the Lonza Nucleofector 2b and program T-016. The electroporated cells were transferred into 2 ml fresh Erythroid Expansion Medium and plated in one well of a 12-well plate and incubated for 2 days at 37°C and 5% CO$_2$.

Day 10: Electroporated cells were collected into a 15-ml conical tube and centrifuged at $200 \times g$ for 5 min. The cell pellet was resuspended in 1 ml DMEM (Life Technologies, 41965039) supplemented with 10% FCS (Life Technologies, 10270-106) and transferred to one well of a 12-well plate, coated with vitronectin (Life Technologies, A14700). The sealed plate was centrifuged at $200 \times g$ for 30 min at 25°C and incubated at 37°C, 5% CO$_2$.

Day 11: The medium was carefully removed from the culture and collected in a 15-ml conical tube, and 0.5 ml E8 medium (Life Technologies, A1517001) supplemented with 0.25 mM n-Butyric Acid Sodium Salt (NaB) (Sigma, B-5887; MW 110.1) were added to each well. The tubes were centrifuged at $300 \times g$ for 5 min, the supernatant was carefully removed and the cell pellet was resuspended in 0.5 ml E8 medium supplemented with 0.25 mM NaB and plated back to the same well so that there was 1 ml of E8 medium supplemented with 0.25 mM NaB per well.

Day 13: The medium was replaced with fresh E8 medium supplemented with 0.25 mM NaB and changed every second day until large colonies of iPSCs become visible. The absence of genomic rearrangements (> 0.2 Mb) of the iPSCs reprogrammed from PBMCs was confirmed by Copy Number Variation analysis (Danecek et al, 2016) performed on genotype data obtained by genome-wide SNP array (Infinium Global Screening Array-24 v1.0; Illumina).

## Generation and analysis of hCOs

iPSCs reprogrammed from human newborn foreskin fibroblasts (CRL-2522, ATCC) (O'Neill et al, 2018; Klaus et al, 2019) were used to generate hCOs (f-CTRL) as previously described (Lancaster et al, 2013; Lancaster & Knoblich, 2015). Organoids were kept in 10-cm dishes on a shaker at 37°C, 5% CO$_2$ and ambient oxygen level with medium changes every 3–4 days. Organoids were analyzed at different days after the initial plating of the cells as indicated in the figures presented.

For electroporation, hCOs were kept in antibiotic-free conditions. Electroporation was performed in hCOs at d35 and fixed 5 days post-electroporation. During the electroporation, hCOs were placed in an electroporation chamber (Harvard Apparatus, Holliston, MA, USA) under a stereoscope, and using a glass microcapillary, 1–2 μl of plasmid DNAs was injected together with Fast Green (0.1%; Sigma) into different ventricles of the hCOs. The hCOs were subsequently electroporated with five pulses applied at 80 V for 50 ms each at intervals of 500 ms (ECM830, Harvard Apparatus). Following electroporation, hCOs were kept for an additional 24 h in antibiotic-free media and then changed into the normal media until fixation. The hCOs were fixed using 4% PFA for 1 h at 4°C, cryopreserved with 30% sucrose, and stored at −20°C. For IHC, 16-μm

cryosections were prepared. For each experiment, many independent ventricles per hCO from at least three different hCOs generated in two independent batches were analyzed.

## Exchange of conditioned medium from b-CTRL and UL4 hCOs

Conditioned medium (CM) was exchanged between b-CTRL and UL4 hCO cultures starting from day 7 of the protocol for hCO generation. CM was switched daily from b-CTRL to UL4 hCOs and vice versa. b-CTRL and UL4 hCOs cultured without CM exchange were used as controls. CM is composed by half fresh medium and half from organoids in culture. This procedure was followed for 29 days.

At day 36, hCOs were collected for FACS analysis according to the procedure described below. Four samples were analyzed for each condition; every sample was a pool of three hCOs.

## Generation of GFP-labeled iPSC line

The GFP-labeled iPSC line was generated using the piggyBac transposase (1 μg) and PB-GFP (1 μg) nucleofection (Chen & LoTurco, 2012). Single cells of iPSCs (f-CTRL) were transfected with the Amaxa Nucleofector 2b (program B-016). GFP colonies were picked and cultured on Matrigel (Corning/VWR International, 354234)-coated plates in mTeSR1 basic medium (Stem Cell Technologies, 85850) supplemented with 1× mTeSR1 supplement (Stem Cell Technologies, 85850) at 37°C and 5% $CO_2$.

## Generation and analysis of fused organoids

Fused hCOs were generated as previously described (Bagley *et al*, 2017). Embryoid bodies (EBs) generated from GFP-labeled f-CTRL iPSCs were patterned in order to have ventral identity, while non-labeled f-CTRL and UL1 iPSCs were patterned in order to have dorsal identity. The fusion of ventralized EBs together with dorsalized ones creates the fused organoids. Briefly, iPSCs were dissociated using Accutase (Sigma-Aldrich, A6964) and approximately 9,000 cells were transferred to one well of an ultra-low-attachment 96-well plate (Corning). Five days later, during the neuronal induction, in order to induce brain regional patterning, EBs from GFP-labeled iPSCs were treated individually with SAG (1:10,000) (Millipore, 566660) + IWP-2 (1:2,000) (Sigma-Aldrich, I0536) for ventral identity and EBs from f-CTRL or UL1 iPSCs with cyclopamine A (1:500) (Calbiochem, 239803) for dorsal identity. After 7 days, one ventral EB and one dorsal EB were embedded together into the same Matrigel (Corning/VWR International, 354234) droplet in order to form a fused organoid. After this point, the generation of organoids followed methods according to (Lancaster & Knoblich, 2015).

## Dissociation of hCOs to generate 2D cultures

The d30 and d60 hCOs generated from b-CTRL, UL1, and UL4 iPSCs, were dissociated to single cells using Accutase (Sigma-Aldrich, A6964). Single cells were then plated onto Polyornithine (10 μg/ml) (Sigma-Aldrich, P3655)/Laminin (10 μg/ml) (Sigma-Aldrich, L2020)-coated coverslips (according to Boyer *et al*, 2012 with some modifications) (Boyer *et al*, 2012) in wells of 24-well plates (Corning). Cells coming from a pool of two to three organoids were plated in six wells of 24-well plates with neural differentiation medium (Lancaster & Knoblich, 2015). After 3 days in culture, cells were fixed with 4% PFA, and then, immunohistochemical analysis was performed using Tubulin Beta 3 (TUBB3) to evaluate the number of neurons and KI67 to evaluate the number of proliferating cells derived from the organoids.

## Mice

All the animals used in this research were kept in the animal facility of the Max Planck Institute of Psychiatry, Munich. All the experimental procedures were performed in accordance with German and European Union guidelines. In this study, the C57BL/6J mouse line was used for the evaluation of the effects of Cstb and R68X mutant overexpression on cell proliferation while Tg(Gad2-EGFP)DJ31Gsat (GAD65-GFP) transgenic mouse strain (López-Bendito *et al*, 2004) and Gad1$^{tm1.1Tama}$ (GAD67-GFP) knock-in mouse strain (Tamamaki *et al*, 2003) were used to evaluate interneurons migration. The day of vaginal plug was considered as embryonic day 0 (E0). All animals used for *in utero* electroporation were female aged between 4 and 6 months.

## Constructs for Cstb or R68X mutant overexpression

To overexpress Cstb or R68X mutant, we used the plasmid constructs previously described (Cipollini *et al*, 2008). Cstb wt protein sequence or R68X mutant was fused to GFP in pEGFP-C1 plasmid, and the empty vector was used as control for electroporations in Figs 2A, C, F, G, EV2D, EV3L, and 4E, H, and L. Cstb wt protein sequence or R68X mutant was fused to HA epitope in pRK7-HA vector (Cipollini *et al*, 2008) and the empty vector was used as control for electroporations in Figs 3D and F, 4N, and EV3G. When HA constructs were used the pmCherry-C1 plasmid (Clontech) was co-electroporated to identify the electroporated regions.

## *In utero* electroporation

As approved by the Government of Upper Bavaria under license number 55.2-1-54-2532-79-2016, pregnant C57BL/6, GAD65-GFP, and GAD67-GFP mice were used. They were weighed and anesthetized by intraperitoneal injection of saline solution containing fentanyl (0.05 mg/kg body weight), midazolam (5 mg/kg body weight), and medetomidine (0.5 mg/kg body weight) (Btm license number 4518395), and embryos were electroporated at E14 as described (Saito, 2006). Plasmids were mixed with Fast Green (2.5 mg/μl; Sigma) and injected at a final concentration of 1 μg/μl. Anesthesia was terminated by injection of buprenorphine (0.1 mg/kg body weight), atipamezole (2.5 mg/kg body weight), and flumazenil (0.5 mg/kg body weight). Brains were fixed 2 or 3 dpe in 4% PFA for 4 h. For immunohistochemistry, 25-μm sections were prepared using a cryotome. For each experiment, at least three different mouse brains per condition were analyzed.

## BrdU injections

Mice received a single intraperitoneal BrdU injection (50 mg/kg body weight) and were sacrificed 30 min later. Tissue preparation and processing were performed as described above.

**Primary cell cultures from mouse embryonic cerebral cortex**

Brains were isolated from embryos of E14 timed pregnant C57BL/6 wild-type mice according to Stahl et al (2013). Briefly, cortices were dissected and pooled in HBSS buffer with 10 mM HEPES and cortical cells were enzymatically dissociated using 0.05% trypsin/EDTA for 15 min at 37°C with subsequent resuspension of cells in DMEM containing 10% fetal calf serum (FCS) to stop the enzymatic reaction. Then, a fire-polished Pasteur pipette, preincubated in DMEM/FCS to coat the glass was used to mechanically dissociate the cells.

Nocodazole treatment: Primary cortical cells were plated at a density of $4 \times 10^5$ cells/well (of a 24-well plate) in DMEM+B27 1:50 and penicillin/streptomycin. The medium was replaced after 2 days in vitro and at 5 days after plating the cells, adding 100 ng/ml nocodazole (Sigma-Aldrich) to the medium. After 4 h, CM was recovered, filtered on 0.22-μm membranes (Millipore) and 10% of the recovered CM was analyzed by WB.

Transfections with shRNA: Dissociated cells were transfected with the Amaxa Nucleofector. The plasmids expressing short hairpins for kinesins were a gift from Tsai et al (2010a, b). shRNAs were inserted into the pRNAT-U6.1/Neo vector (Genscript):

shRNA Kif5a: GGTGCTGAATGGACTGATGAA
shRNA Kif1a: TCCAGAACTTGACTCCAAGAA.

At 4 days post-transfection, CM was recovered and filtered on 0.22-μm membranes (Millipore) and 10% of the recovered CM was analyzed by WB.

Transfection for overexpression of Cstb and R68X mutant: dissociated cells were transfected with the Amaxa Nucleofector. pEGFP-C1 plasmid as a control or pEGFP-Cstb or pEGFP-R68X was transfected. At 4 days post-transfection, cells were lysed in RIPA buffer and CM was recovered, filtered on 0.22-μm membranes (Millipore) and 10% of the recovered CM was analyzed by WB.

**Immunohistochemistry**

For IHC, sections were post-fixed using 4% PFA for 10 min and permeabilized with 0.3% Triton for 5 min. After post-fixation and permeabilization, sections were blocked with 0.1% Tween, 10% Normal Goat Serum (Biozol, VEC-S-1000). Primary and secondary antibodies were diluted in blocking solution. Nuclei were visualized using 0.5 mg/ml 4,6-diamidino-2-phenylindole (DAPI) (Sigma-Aldrich, D9542). Immunostained sections were analyzed using a Leica laser-scanning microscope.

Notably, for CSTB antibody, we carried out an extra pretreatment for antigen retrieval before the post-fixation step: We incubated the section in a freshly made 10 mM citric buffer (pH 6) in a microwave for 1 min at 720 W and for 10 min at 120 W and then left it to cool down for 20 min at RT.

Prior to BrdU immunostaining, DNA was denatured by 30-min incubation in 4 M HCl, followed by neutralization for 10 min in 0.1 M sodium borate at pH 8.5. Two 5-min washing steps were performed before continuing with the immunohistochemical procedure.

| Antigen | Dilution | Vendor | Catalog no. |
|---|---|---|---|
| CSTB | 1:500 | Antikoerper | AbIN271833 |
| ACETYLATED TUBULIN | 1:6,000 | Sigma-Aldrich | T7451 |
| PH3 | 1:500 | Millipore | 06-570 |
| GAPDH | 1:6,000 | Millipore | CB1001 |
| GFP | 1:1,000 | Aves Lab | GFP-1020 |
| RFP | 1:1,000 | Rockland | 600-901-379 |
| DCX | 1:2,000 | Millipore | AB2253 |
| TUBB3 | 1:500 | Sigma-Aldrich | T8660 |
| KI67 (mouse Ab) | 1:500 | DAKO | M7248 |
| KI67 (rabbit Ab) | 1:500 | Abcam | AB15580 |
| ACTIN (Actb) | 1:1,000 | Merck Millipore | MAB1501 |
| BrdU | 1:200 | Abcam | AB6326 |
| PAX6 | 1:500 | Abcam | AB78545 |
| Ccnd1 | 1:100 | Thermo Scientific | RM-9104-S0 |
| TBR2 | 1:500 | Merck Millipore | Ab2283 |

**CSF sample collection**

Samples of CSF were collected from the fourth ventricle using microcapillary samplers (E14, typically 1–2 μl) (Johansson et al, 2006). All CSF samples were microscopically examined for traces of blood over a white background and samples that showed any sign of blood contamination were discarded.

**Western blot**

iPSCs and tissues from mouse cortices were lysed in RIPA buffer with protease and phosphatase inhibitors (Roche, Basel, Switzerland) and proteins were separated by SDS–PAGE with a 12.5% acrylamide gel. 50 mM DTT was added to some protein extracts (Fig EV4C) in order to dissociate protein aggregates and to verify that CSTB (MW 11 kDa) was absent and not forming aggregates. GAPDH (MW 37 kDa) and Actb (MW 42 kDa) immunostaining were used as loading controls. CM from hCO cultures was collected and concentrated by using Amicon® Ultra-15 Centrifugal Filter Units according to the instructions (Millipore, UFC901024). We took care of collecting the same volume of CM from wells or from 10-cm dishes with comparable numbers of cells or hCOs in culture. Proteins were transferred to a nitrocellulose membrane (GE Healthcare, Chalfont St Giles, Buckinghamshire, Great Britain). After blotting, membranes were shortly colored with a Ponceau Red solution (Serva, 33427.01) in order to verify the quality of the transfer. For detection, membranes were incubated with primary antibodies overnight, then with horseradish peroxidase-labeled secondary antibodies at room temperature, and afterward treated with ECL Western Blotting Detection Solution (Millipore, Billerica, MA, USA) to visualize bands. Bands were quantified using ImageJ software.

**FACS analysis: intracellular FACS and nuclei isolation**

For intracellular FACS analysis, hCOs (at 60 days in culture) were collected. Three to four samples were analyzed as biological replicates;

every sample contained three individual hCOs. The procedure followed was described previously (Barbosa *et al*, 2016; Di Giaimo *et al*, 2019; Klaus *et al*, 2019) with the difference that DCX$^+$ cells were resuspended in staining solution containing Alexa Fluor 488 anti-guinea pig. For the CM exchange protocol, Ki67 (1:200; marca) and PH3 (1:300; marca) stainings were used to detect proliferating cells and as secondary antibodies Alexa Fluor 488 anti-rabbit was used.

For nuclei isolation from d135 organoids, we followed the methods according to Krishnaswami *et al* (2016), with small modifications. Briefly, hCOs were dissociated by Dounce homogenization and filtered with BD Falcon tubes with a cell strainer cap (352235, Corning). RNase inhibitor (M0314, NEB) (0.4 U/µl) and DNase I (M0303, NEB) (1 U/µl) were added to the homogenization buffer. RNase inhibitor (0.2 U/µl) was added to the blocking solution. Primary antibodies against Pax6 (PRB-278p; Biolegend) and NeuN (MAB377; Millipore) were used at 1:2,500 dilution and 1:1,000, respectively. Secondary antibodies goat anti-rabbit Alexa Fluor 488 (A-11008; Thermo Scientific) and Alexa Fluor® 546 Goat Anti-Mouse IgG1 (γ1) (A-21123; Thermo Scientific) were used at 1:2,500 dilution. DAPI (D9542) (0.5 mg/ml) was added during the last 10 min of the secondary antibody incubation. Only a fraction of hCO homogenate was incubated with secondary antibodies and analyzed as controls for setting the FACS gates.

FACS analysis was performed with a FACS Melody (BD) in BD FACS Flow TM medium, with a nozzle diameter of 100 µm. Debris and aggregated cells were gated out by forward scatter, sideward scatter; single cells were gated out by FSC-W/FSC-A. Gating for fluorophores was done using samples stained with secondary antibody only.

## Generation and validation of microRNAs targeting the *Cstb* gene

MicroRNAs (miRNAs) targeting mouse *Cstb* gene was generated using the BLOCK-iT system from Invitrogen (Thermo Fisher). miRNA sequences were determined using Invitrogen's RNAi design tool. *Cstb*-targeting miRNA oligonucleotides were annealed and ligated into a GFP-containing entry vector pENTR-GW/EmGFP-miR containing the miRNA backbones using T4 DNA Ligase (Thermo Fisher). Afterward, the miRNA sequences (plus GFP) were cloned into the pCAG-GS destination vector using the Gateway system (Thermo Fisher) according to the manufacturer's recommendations. The resulting miRNA expression plasmid was sequenced. The knockdown efficiency of the miRNA was tested via qPCR and was validated by immunohistochemistry on primary E14 cortex cells (Stahl *et al*, 2013). miRNA-Cstb was then used for neural progenitor transfection in the developing mouse cortex by *in utero* electroporation. The following constructs were chosen. A miRNA with scrambled sequence ("miRneg") was used as negative control

| Oligo name | Sequence (5′–3′) |
|---|---|
| miRNA-Cstb top | TGCTGTTTCGATTCAAGCTGGGACTTGTTTTGGCCACTGACTGACAAGTCCCATTGAATCGAAA |
| miRNA-Cstb bottom | CCTGTTTCGATTCAATGGGACTTGTCAGTCAGTGGCCAAAACAAGTCCCAGCTTGAATCGAAAC |
| miRneg top | TGCTGAAATGTACTGCGCCGTGGAGACGTTTTGGCCACTGACTGACTTCTCCACGCAGTATTTCAGG |
| miRneg bottom | CCTGAAATGTACTGCGTGGAGACGTCAGTCAGTGGCCAAAACGTCTCCACGCGCAGTATTTCAGCA |

## RNA extraction, cDNA synthesis, and real-time qPCR

iPSCs and NEUN$^+$ and PAX6$^+$ nuclei from FACS sorting were lysed in QIAzol® Lysis Reagent (Qiagen, 79306). RNA was extracted using RNA Clean & Concentrator Kit (Zymo Research, R1015) and cDNA was synthesized using SuperScript III reverse transcriptase (Thermo Fisher, 18080-044) with Random primers (Invitrogen, 48190-11) according to the manufacturer's protocol. Subsequently, qPCR was performed in triplicates on a LightCycler® 480 II (Roche) using the LightCycler® 480 SYBR Green I Master (#04707516001, Roche).

The primers sequences were as follows:

*CSTB:* FW: 5′-CTGTGTTTAAGGCCGTGTCA-3′; RV: 5′-AGGTCAGCTCATCATGCTTG-3′ (Wang *et al*, 2014)
*GAPDH:* FW: 5′-AATCCCATCACCATCTTCCAGGA-3′; RV: 5′-TGGACTCCACGACGTACTCAG-3′
*PAX6*: FW: 5′-ACCCATTATCCAGATGTGTTTGC-3′; RV: 5′-ATGGTGAAGCTGGGCATAGG-3′ (Li *et al*, 2017)
*NEUN*: FW: 5′-CCAAGCGGCTACACGTCT-3′; RV: 5′-GCTCGGTCAGCATCTGAG-3′ (Giorgetti *et al*, 2012)
*NESTIN*: FW: 5′-GGGAAGAGGTGATGGAACCA-3′; RV: 5′-AAGCCCTGAACCCTCTTTGC-3′ (Giorgetti *et al*, 2012).

The qPCR was done with initial denaturation at 95°C for 10 min, followed by 45 cycles of 95°C for 10 s, 60°C for 10 s, and 72°C for 10 s. Relative expression was calculated using the $\Delta\Delta C_p$ method.

## Quantifications and statistical analyses

Statistical analysis and plotting of data were performed with GraphPad Prism® version 7.04. Statistical significance between unpaired groups was analyzed using *t*-test. Statistical significance between two unpaired groups was analyzed using Mann–Whitney test as indicated in the figure legends. All experiments were reproduced at least three times independently and all attempts at replication were successful. No randomization was performed, but different batches of cerebral organoids and different animals for each experiment were used (at least $n = 3$). Analysis was not performed blindly, but all acquired data were verified by a second investigator.

## Sample preparation for proteomic analyses

A pool of 20–30 different b-CTRL or UL1 hCOs were lysed in RIPA buffer (50 mM Tris–HCl pH 8.8, 150 mM NaCl, 0.1% SDS, 0.5% NP-40, 0.5% DOC; protease and phosphatase inhibitor cocktail; Sigma-Aldrich). Protein concentration was determined by Bradford assay (Bio-Rad). An aliquot of 100 µg of proteins was used for mass spectrometry analysis. Proteins were diluted in triethylammonium bicarbonate buffer (TEAB, 100 mM final concentration) containing 1% SDS. The protein concentration was then determined by using the Pierce BCA Protein assay kit (Thermo Scientific, Rockford, Illinois, USA). For each condition, equal amounts of proteins (80 µg in 100 µl of 100 mM TEAB) were digested with trypsin and labeled with the following TMT isobaric tags according to the procedure described elsewhere (Napolitano *et al*, 2019; Russo *et al*, 2019): 129N and 127C for UL1 and b-CTRL samples, respectively. TMT-labeled samples were then mixed and diluted in

2% TFA to a final concentration of 0.5 µg/µl for LC-MS/MS analyses.

### High-resolution nanoLC–Tandem mass spectrometry

Aliquots of TMT-labeled peptides (2.5 µg) were analyzed by high-resolution nanoLC–tandem mass spectrometry using a Q-Exactive Orbitrap mass spectrometer equipped with an EASY-Spray nanoelectrospray ion source (Thermo Scientific, Rockford, Illinois, USA), coupled to a Thermo Scientific Dionex UltiMate 3000 RSLCnano system (Thermo Scientific, Rockford, Illinois, USA) as reported elsewhere (Russo *et al*, 2019).

### Protein identification and quantification

The acquired raw files were analyzed with the Thermo Scientific Proteome Discoverer 2.1 software (Thermo Scientific, Rockford, Illinois, USA) using the SEQUEST HT search engine. The HCD MS/MS spectra were searched against the Homo sapiens Uniprot_sprot database (release 2019_11, 20380 entries) by using the following parameters: trypsin (Full) as digestion enzyme with two missed cleavage sites; mass tolerances: 10 ppm and 0.02 Da, for precursor and fragment ions, respectively; dynamic modification: methionine oxidation (+15.995 Da); static modifications: carbamidomethylation of cysteine (+57.021 Da) and the TMT label on lysines and the N-terminus (229.1629). False discovery rates (FDRs) for peptide spectral matches (PSMs) were calculated and filtered using the Percolator node in Proteome Discoverer that was run with the following settings: Maximum Delta Cn 0.05, a strict target FDR of 0.01, a relaxed target FDR of 0.05, and validation based on q-value. Protein identifications were accepted when the protein FDR was below 1% and when present in at least two out of three replicate injections with at least two peptides.

### Bioinformatic analyses

For bioinformatic analyses, proteins with $\log_2$ fold change values ($\log_2$FC) $\geq 0.3$ and $\leq -0.3$ were considered as differentially expressed (DE). The enriched Gene Ontology (GO) Biological Processes for DE proteins terms were extracted by using the g:GOSt tool of the g:Profiler toolset (https://biit.cs.ut.ee/gprofiler/gost). g:SCS-adjusted *P*-values were considered significant if $< 0.05$. GOChord plots of selected GO categories were drawn through the GOplot package v 1.0.2 of the RStudio v 1.2.1335 environment for R (http://www.R-project.org). Volcano plot of DE proteins was also drawn in RStudio.

## Data availability

The datasets produced in this study are available in the following databases: MS-based proteomic data: PRIDE database PXD018021 (https://www.ebi.ac.uk/pride/archive/projects/PXD018021).

**Expanded View** for this article is available online.

## Acknowledgements

We thank the families participating in this study for their involvement. We thank Pia Johannsson for help collecting CSF; Barbara Wölfel, Vincenza Sportelli, Timucin Öztürk, and Andrea Steiner for their excellent technical support; Richard Vallee and Joseph Lo Turco for sharing plasmids; Lucia Trastulla for supporting CNV analysis; Davide Piccolo for help with qPCR analyses; Marialuisa Melli and Jessica Keverne for critical reading of the manuscript and Isabel Buchsbaum for critical reading of the manuscript and for generating schematic drawings; Guillermina Lopez-Bendito and Nobuaki Tamamaki for sharing mouse lines; and Jovica Ninkovic for initial suggestions and support. We also thank ERASMUS+ traineeship program, Key Action 103: Learning Mobility for Individuals within Europe, for supporting student exchange between University of Naples Federico II and Max Planck Institute of Psychiatry. This work was supported by funding from the Max Planck Society and by "Finanziamento Ricerca di Ateneo" from University of Naples Federico II. Fabrizia Pipicelli is supported by DFG (CA 1205/4-1), and Francesco Di Matteo is supported by ERA-Net Neuron (nEUrotalk). Christina Kyrousi is supported by ERA-Net E-Rare (HETER-OMICS).

## Author contributions

SC and RDG conceived the project. RDG, SC, FDM, and FP designed the experiments. RDG, SC, FDM, FP, CK, IT, ACA-M, MGi, and AH performed the experiments. AC and RR performed proteomic and MS data analyses. EP and MC prepared samples for proteomic analysis and validated MS results. LC and EC provided patients' PBMCs. MGö was involved in the start of the project and contributed to data discussion. SC and RDG wrote the manuscript. All authors provided ongoing critical review of results and commented on the manuscript.

## Conflict of interest

The authors declare that they have no conflict of interest.

## References

Aka S, Alanay Y, Boodhansingh KE, Stanley CA, Semiz S (2016) Seizures and diagnostic difficulties in hyperinsulinism-hyperammonemia syndrome. *Turk J Pediatr* 58: 541–544

### The paper explained

#### Problem
Unverricht–Lundborg-type (EPM1) epilepsy is a neurodegenerative disorder with the highest incidence of progressive myoclonus epilepsies worldwide. Mutations in cystatin B (CSTB) are the primary genetic cause of EPM1. However, the molecular and cellular mechanisms involved in the development of EPM1 are not fully understood.

#### Results
We showed that CSTB is essential for cell proliferation and migration during brain development. We demonstrated that CSTB is secreted and modulates the extracellular environment. Perturbations of the extracellular environment can lead to an imbalance of excitation/inhibition, contributing to the insurgence of several neurological and psychiatric disorders.

#### Impact
Overall, our study reveals a role for CSTB in regulating the early stages of human neurogenesis by acting as a short- and long-distance mediator and sheds new light on the possible mechanism underlying this form of epilepsy suggesting possible therapeutic strategies to antagonize the progression of the pathological phenotype.

Alakurtti K, Weber E, Rinne R, Theil G, De Haan GJ, Lindhout D, Salmikangas P, Saukko P, Lahtinen U, Lehesjoki AE (2005) Loss of lysosomal association of cystain B proteins representing progressive myoclonus epilepsy, EPM1, mutations. *Eur J Hum Genet* 13: 208−215

Bagley JA, Reumann D, Bian S, Lévi-Strauss J, Knoblich JA (2017) Fused cerebral organoids model interactions between brain regions. *Nat Methods* 14: 743−751

Barbosa JS, Di Giaimo R, Götz M, Ninkovic J (2016) Single-cell *in vivo* imaging of adult neural stem cells in the zebrafish telencephalon. *Nat Protoc* 11: 1360−1370

Bassuk AG, Wallace RH, Buhr A, Buller AR, Afawi Z, Shimojo M, Miyata S, Chen S, Gonzalez- Alegre P, Griesbach HL *et al* (2008) A homozygous mutation in human PRICKLE1 causes an autosomal-recessive progressive myoclonus epilepsy-ataxia syndrome. *Am J Hum Genet* 83: 572−581

Berkovic SF, Andermann F, Carpenter S, Wolfe LS (1986) Progressive myoclonus epilepsies: specific causes and diagnosis. *N Engl J Med* 315: 296−305

Boyer LF, Campbell B, Larkin S, Mu Y, Gage FH (2012) Dopaminergic differentiation of human pluripotent cells. *Curr Protoc Stem Cell Biol* Chapter 1: Unit1H.6

Butinar M, Prebanda MT, Rajković J, Jerič B, Stoka V, Peters C, Reinheckel T, Krüger A, Turk V, Turk B *et al* (2014) Stefin B deficiency reduces tumor growth via sensitization of tumor cells to oxidative stress in a breast cancer model. *Oncogene* 33: 3392−3400

Buzzi A, Chikhladze M, Falcicchia C, Paradiso B, Lanza G, Soukupova M, Marti M, Morari M, Franceschetti S, Simonato M (2012) Loss of cortical GABA terminals in Unverricht-Lundborg disease. *Neurobiol Dis* 47: 216−224

Calkins CC, Sameni M, Koblinski J, Sloane BF, Moin K (1998) Differential localization of cysteine protease inhibitors and a target cysteine protease, cathepsin B, by immuno-confocal microscopy. *J Histochem Cytochem* 46: 745−751

Canafoglia L, Gennaro E, Capovilla G, Gobbi G, Boni A, Beccaria F, Viri M, Michelucci R, Agazzi P, Assereto S *et al* (2012) Electroclinical presentation and genotype-phenotype relationships in patients with Unverricht-Lundborg disease carrying compound heterozygous CSTB point and indel mutations. *Epilepsia* 53: 2120−2127

Cava C, Manna I, Gambardella A, Bertoli G, Castiglioni I (2018) Potential role of miRNAs as theranostic biomarkers of epilepsy. *Mol Ther Nucleic Acids* 13: 275−290

Ceru S, Konjar S, Maher K, Repnik U, Krizaj I, Bencina M, Renko M, Nepveu A, Zerovnik E, Turk B *et al* (2010) Stefin B interacts with histones and cathepsin L in the nucleus. *J Biol Chem* 285: 10078−10086

Chen F, Loturco J (2012) A method for stable transgenesis of radial glia lineage in rat neocortex by piggyBac mediated transposition. *J Neurosci Methods* 207: 172−180

Chou B-K, Gu H, Gao Y, Dowey SN, Wang Y, Shi J, Li Y, Ye Z, Cheng T, Cheng L (2015) A facile method to establish human induced pluripotent stem cells from adult blood cells under feeder-free and xeno-free culture conditions: a clinically compliant approach. *Stem Cells Transl Med* 4: 320−332

Cipollini E, Riccio M, Di Giaimo R, Dal Piaz F, Pulice G, Catania S, Caldarelli I, Dembic M, Santi S, Melli M (2008) Cystatin B and its EPM1 mutants are polymeric and aggregate prone *in vivo*. *Biochim Biophys Acta Mol Cell Res* 1783: 312−322

D'amato E, Kokaia Z, Nanobashvili A, Reeben M, Lehesjoki AE, Saarma M, Lindvall O (2000) Seizures induce widespread upregulation of cystatin B, the gene mutated in progressive myoclonus epilepsy, in rat forebrain neurons. *Eur J Neurosci* 12: 1687−1695

Danecek P, McCarthy SA, Consortium HS, Durbin R (2016) A method for checking genomic integrity in cultured cell lines from snp genotyping data. *PLoS ONE* 11: e0155014

Danner N, Julkunen P, Hyppönen J, Niskanen E, Säisänen L, Könönen M, Koskenkorva P, Vanninen R, Kälviäinen R, Mervaala E (2013) Alterations of motor cortical excitability and anatomy in unverricht-lundborg disease. *Mov Disord* 28: 1860−1867

Di Giaimo R, Aschenbroich S, Ninkovic J (2019) Fluorescence-activated cell sorting-based isolation and characterization of neural stem cells from the adult zebrafish telencephalon. *Methods Mol Biol* 1938: 49−66

Di Giaimo R, Riccio M, Santi S, Galeotti C, Ambrosetti DC, Melli M (2002) New insights into the molecular basis of progressive myoclonus epilepsy: a multiprotein complex with cystatin B. *Hum Mol Genet* 11: 2941−2950

Dibbens LM, Michelucci R, Gambardella A, Andermann F, Rubboli G, Bayly MA, Joensuu T, Vears DF, Franceschetti S, Canafoglia L *et al* (2009) SCARB2 mutations in progressive myoclonus epilepsy (PME) without renal failure. *Ann Neurol* 66: 532−536

Dityatev A (2010) Remodeling of extracellular matrix and epileptogenesis. *Epilepsia* 1: 61−65

Dowey SN, Huang X, Chou BK, Ye Z, Cheng L (2012) Generation of integration-free human induced pluripotent stem cells from postnatal blood mononuclear cells by plasmid vector expression. *Nat Protoc* 7: 2013−2021

Eldridge R, Stern R, Anainen MI, Koerber T, Wilder BJ (1983) 'BALTIC' myoclonus epilepsy: hereditary disorder of childhood made worse by phenytoin. *Lancet* 2: 838−842

Ercegovac M, Jovic N, Sokic D, Savic-Radojevic A, Coric V, Radic T, Nikolic D, Kecmanovic M, Matic M, Simic T *et al* (2015) GSTA1, GSTM1, GSTP1 and GSTT1 polymorphisms in progressive myoclonus epilepsy: A Serbian case-control study. *Seizure* 32: 30−36

Giorgetti A, Marchetto MCN, Li M, Yu D, Fazzina R, Mu Y, Adamo A, Paramonov I, Cardoso JC, Monasterio MB *et al* (2012) Cord blood-derived neuronal cells by ectopic expression of Sox2 and c-Myc. *Proc Natl Acad Sci USA* 109: 12556−12561

Guo W, Shang DM, Cao JH, Feng K, He YC, Jiang Y, Wang SP, Gao YF (2017) Identifying and analyzing novel epilepsy-related genes using random walk with restart algorithm. *Biomed Res Int* 2017: 6132436

Houseweart MK, Vilaythong A, Yin XM, Turk B, Noebels JL, Myers RM (2003) Apoptosis caused by cathepsins does not require Bid signaling in an *in vivo* model of progressive myoclonus epilepsy (EPM1). *Cell Death Differ* 10: 1329−1335

Joensuu T, Lehesjoki AE, Kopra O (2008) Molecular background of EPM1 - Unverricht-Lundborg disease. *Epilepsia* 49: 557−563

Joensuu T, Tegelberg S, Reinmaa E, Segerstråle M, Hakala P, Pehkonen H, Korpi ER, Tyynelä J, Taira T, Hovatta I *et al* (2014) Gene expression alterations in the cerebellum and granule neurons of Cstb-/- mouse are associated with early synaptic changes and inflammation. *PLoS ONE* 9: e89321

Johansson PA, Dziegielewska KM, Ek CJ, Habgood MD, Liddelow SA, Potter AM, Stolp HB, Saunders NR (2006) Blood-CSF barrier function in the rat embryo. *Eur J Neurosci* 24: 65−76

Kälviäinen R, Khyuppenen J, Koskenkorva P, Eriksson K, Vanninen R, Mervaala E (2008) Clinical picture of EPM1-Unverricht-Lundborg disease. *Epilepsia* 49: 549−556

Kaur G, Mohan P, Pawlik M, Derosa S, Fajiculay J, Che S, Grubb A, Ginsberg SD, Nixon RA, Levy E (2010) Cystatin C rescues degenerating neurons in a cystatin B-knockout mouse model of progressive myoclonus epilepsy. *Am J Pathol* 177: 2256−2267

Klaus J, Kanton S, Kyrousi C, Ayo-Martin AC, Di Giaimo R, Riesenberg S, O'Neill AC, Camp JG, Tocco C, Santel M *et al* (2019) Altered neuronal migratory trajectories in human cerebral organoids derived from individuals with neuronal heterotopia. *Nat Med* 25(4): 561–568

Koskenkorva P, Khyuppenen J, Niskanen E, Könönen M, Bendel P, Mervaala E, Lehesjoki AE, Kälviäinen R, Vanninen R (2009) Motor cortex and thalamic atrophy in Unverricht-Lundborg disease: voxel-based morphometric study. *Neurology* 73: 606–611

Koskiniemi M, Donner M, Majuri H, Haltia M, Norio R (1974) Progressive myoclonus epilepsy: a clinical and histopatliological study. *Acta Neurol Scand* 50: 307–332

Krishnaswami SR, Grindberg RV, Novotny M, Venepally P, Lacar B, Bhutani K, Linker SB, Pham S, Erwin JA, Miller JA *et al* (2016) Using single nuclei for RNA-seq to capture the transcriptome of postmortem neurons. *Nat Protoc* 11: 499–524

Lafrenière RG, Rochefort DL, Chrétien N, Rommens JM, Cochius JI, Kälviäinen R, Nousiainen U, Patry G, Farrell K, Söderfeldt B *et al* (1997) Unstable insertion in the 5' flanking region of the cystatin B gene is the most common mutation in progressive myoclonus epilepsy type 1, EPM1. *Nat Genet* 15: 298–302

Lalioti MD, Scott HS, Buresi C, Rossier C, Bottani A, Morris MA, Malafosse A, Antonarakis SE (1997) Dodecamer repeat expansion in cystatin B gene in progressive myoclonus epilepsy. *Nature* 386: 847–851

Lancaster MA, Renner M, Martin CA, Wenzel D, Bicknell LS, Hurles ME, Homfray T, Penninger JM, Jackson AP, Knoblich JA (2013) Cerebral organoids model human brain development and microcephaly. *Nature* 501: 373–379

Lancaster MA, Knoblich J (2015) Generation of cerebral organoids from human pluripotent stem. *Nat Protoc* 9: 2329–2340

Lehtinen MK, Tegelberg S, Schipper H, Su H, Zukor H, Manninen O, Kopra O, Joensuu T, Hakala P, Bonni A *et al* (2009) Cystatin B deficiency sensitizes neurons to oxidative stress in progressive myoclonus epilepsy, EPM1. *J Neurosci* 29: 5910–5915

Li Y, Muffat J, Omer A, Bosch I, Lancaster MA, Sur M, Gehrke L, Knoblich JA, Jaenisch R (2017) Induction of expansion and folding in human cerebral organoids. *Cell Stem Cell* 20: 385–396.e3

Lim BC, Matsumoto S, Yamamoto H, Mizuno H, Kikuta J, Ishii M, Kikuchi A (2016) Prickle1 promotes focal adhesion disassembly in cooperation with the CLASP-LL5β complex in migrating cells. *J Cell Sci* 129: 3115–3129

Liu C, Lin C, Whitaker DT, Bakeri H, Bulgakov OV, Liu P, Lei J, Dong L, Li T, Swaroop A (2013) Prickle1 is expressed in distinct cell populations of the central nervous system and contributes to neuronal morphogenesis. *Hum Mol Genet* 22: 2234–2246

López-Bendito G, Sturgess K, Erdélyi F, Szabó G, Molnár Z, Paulsen O (2004) Preferential origin and layer destination of GAD65-GFP cortical interneurons. *Cereb Cortex* 14: 1122–1133

Mancini GMS, Schot R, De Wit MCY, De Coo RF, Oostenbrink R, Bindels-De Heus K, Berger LPV, Lequin MH, De Vries FAT, Wilke M *et al* (2016) CSTB null mutation associated with microcephaly, early developmental delay, and severe dyskinesia. *Neurology* 86: 877–878

Marín O (2012) Interneuron dysfunction in psychiatric disorders. *Nat Rev Neurosci* 13: 107–120

Napolitano F, De Rosa A, Russo R, Di Maio A, Garofalo M, Federici M, Migliarini S, Ledonne A, Rizzo FR, Avallone L *et al* (2019) The striatal-enriched protein Rhes is a critical modulator of cocaine-induced molecular and behavioral responses. *Sci Rep* 9: 15294

Nigri A, Visani E, Bertolino N, Nanetti L, Mariotti C, Panzeri M, Bruzzone MG, Franceschetti S, Canafoglia L (2017) Cerebellar involvement in patients with mild to moderate myoclonus due to EPM1: structural and functional MRI findings in comparison with healthy controls and ataxic patients. *Brain Topogr* 30: 380–389

O'Neill AC, Kyrousi C, Einsiedler M, Burtscher I, Drukker M, Markie DM, Kirk EP, Götz M, Robertson SP, Cappello S (2018) Mob2 insufficiency disrupts neuronal migration in the developing cortex. *Front Cell Neurosci* 12: 57

Parsons JT, Horwitz AR, Schwartz MA (2010) Cell adhesion: integrating cytoskeletal dynamics and cellular tension. *Nat Rev Mol Cell Biol* 11: 633–643

Penna E, Cerciello A, Chambery A, Russo R, Cernilogar FM, Pedone EM, Perrone-Capano C, Cappello S, Di Giaimo R, Crispino M (2019) Cystatin B involvement in synapse physiology of rodent brains and human cerebral organoids. *Front Mol Neurosci* 12: 195

Pennacchio LA, Lehesjoki AE, Stone NE, Willour VL, Virtaneva K, Miao J, D'amato E, Ramirez L, Faham M, Koskiniemi M *et al* (1996) Mutations in the gene encoding cystatin B in progressive myoclonus epilepsy (EPM1). *Science* 271: 1731–1734

Pennacchio LA, Bouley DM, Higgins KM, Scott MP, Noebels JL, Myers RM (1998) Progressive ataxia, myoclonic epilepsy and cerebellar apoptosis in cystatin B-deficient mice. *Nat Genet* 20: 251–258

Perez-Riverol Y, Csordas A, Bai J, Bernal-Llinares M, Hewapathirana S, Kundu DJ, Inuganti A, Griss J, Mayer G, Eisenacher M *et al* (2019) The PRIDE database and related tools and resources in 2019: improving support for quantification data. *Nucleic Acids Res* 47: D442–D450

Polajnar M, Zavašnik-Bergant T, Škerget K, Vizovišek M, Vidmar R, Fonović M, Kopitar-Jerala N, Petrovič U, Navarro S, Ventura S *et al* (2014) Human stefin B role in cell's response to misfolded proteins and autophagy. *PLoS ONE* 9: e102500

Polioudakis D, De La Torre-Ubieta L, Langerman J, Elkins AG, Shi X, Stein JL, Vuong CK, Nichterwitz S, Gevorgian M, Opland CK *et al* (2019) A single-cell transcriptomic atlas of human neocortical development during mid-gestation. *Neuron* 103: 785–801.e8

Reczek D, Schwake M, Schröder J, Hughes H, Blanz J, Jin X, Brondyk W, Van Patten S, Edmunds T, Saftig P (2007) LIMP-2 is a receptor for lysosomal mannose-6-phosphate-independent targeting of β-glucocerebrosidase. *Cell* 131: 770–783

Riccio M, Santi S, Dembic M, Di Giaimo R, Cipollini E, Costantino-Ceccarini E, Ambrosetti D, Maraldi NM, Melli M (2005) Cell-specific expression of the epm1 (cystatin B) gene in developing rat cerebellum. *Neurobiol Dis* 20: 104–114

Rispoli A, Cipollini E, Catania S, Di Giaimo R, Pulice G, Van Houte S, Sparla F, Dal Piaz F, Roncarati D, Trost P *et al* (2013) Insights in progressive myoclonus epilepsy: HSP70 promotes cystatin B polymerization. *Biochim Biophys Acta* 1834: 2591–2599

Rubenstein JLR, Merzenich MM (2003) Model of autism: increased ratio of excitation/inhibition in key neural systems. *Genes Brain Behav* 2: 255–267

Russo R, Matrone N, Belli V, Ciardiello D, Valletta M, Esposito S, Pedone PV, Ciardiello F, Troiani T, Chambery A (2019) Macrophage migration inhibitory factor is a molecular determinant of the anti-EGFR monoclonal antibody cetuximab resistance in human colorectal cancer cells. *Cancers (Basel)* 11: E1430

Saito T (2006) *In vivo* electroporation in the embryonic mouse central nervous system. *Nat Protoc* 1: 1552–1558

Savasta S, Verrotti A, Spartà MV, Foiadelli T, Villa MP, Parisi P (2015) Unilateral periventricular heterotopia and epilepsy in a girl with Ehlers-Danlos syndrome. *Epilepsy Behav Case Rep* 4: 27–29

Shang W, Liu WH, Zhao XH, Sun QJ, Bi JZ, Chi ZF (2008) Expressions of glutathione S-transferase alpha, mu, and pi in brains of medically intractable epileptic patients. *BMC Neurosci* 9: 67

Shannon P, Pennacchio LA, Houseweart MK, Minassian BA, Myers RM (2002) Neuropathological changes in a mouse model of progressive myoclonus epilepsy: Cystatin B deficiency and Unverricht-Lundborg disease. *J Neuropathol Exp Neurol* 61: 1085–1091

Silva CG, Peyre E, Nguyen L (2019) Cell migration promotes dynamic cellular interactions to control cerebral cortex morphogenesis. *Nat Rev Neurosci* 20: 318–329

Stahl R, Walcher T, De Juan Romero C, Pilz GA, Cappello S, Irmler M, Sanz-Aquela JM, Beckers J, Blum R, Borrell V *et al* (2013) Trnp1 regulates expansion and folding of the mammalian cerebral cortex by control of radial glial fate. *Cell* 153: 535–549

Tamamaki N, Yanagawa Y, Tomioka R, Miyazaki JI, Obata K, Kaneko T (2003) Green fluorescent protein expression and colocalization with calretinin, parvalbumin, and somatostatin in the GAD67-GFP knock-in mouse. *J Comp Neurol* 467: 60–79

Tsai JW, Lian WN, Kemal S, Kriegstein AR, Vallee RB (2010a) Kinesin 3 and cytoplasmic dynein mediate interkinetic nuclear migration in neural stem cells. *Nat Neurosci* 13: 1463–1471

Tsai JW, Lian WN, Kemal S, Kriegstein AR, Vallee RB (2010b) An unconventional kinesin and cytoplasmic dynein are responsible for interkinetic nuclear migration in neural stem cells. *Nat Neurosci* 13: 1463–1471

Virtaneva K, D'amato E, Miao J, Koskiniemi M, Norio R, Avanzini G, Franceschetti S, Michelucci R, Tassinari CA, Omer S *et al* (1997) Unstable minisatellite expansion causing recessively inherited myoclonus epilepsy, EPM1. *Nat Genet* 15: 393–396

Wang X, Gui L, Zhang Y, Zhang J, Shi J, Xu G (2014) Cystatin B is a progression marker of human epithelial ovarian tumors mediated by the TGF-β signaling pathway. *Int J Oncol* 44: 1099–1106

Yang H, Ganguly A, Cabral F (2010) Inhibition of cell migration and cell division correlates with distinct effects of microtubule inhibiting drugs. *J Biol Chem* 285: 32242–32250

Žerovnik E (2019) Possible mechanisms by which stefin B could regulate proteostasis and oxidative stress. *Cells* 8: E70

Zhou L, Xiao X, Li S, Jia X, Wang P, Sun W, Zhang F, Li J, Li T, Zhang Q (2018) Phenotypic characterization of patients with early-onset high myopia due to mutations in COL2A1 or COL11A1: why not stickler syndrome? *Mol Vis* 24: 560–573

