## [Review Process File · EMBO Molecular Medicine]

Cystatin B is essential for proliferation and interneuron migration in individuals with EPM1 epilepsy

Francesco Di Matteo, Fabrizia Pipicelli, Christina Kyrousi, Isabella Tovecci, Eduardo Penna, Marianna Crispino, Angela Chambery, Rosita Russo, Ane Cristina Ayo-Martin, Martina Giordano, Anke Hoffmann, Emilio Ciusani, Laura Canafoglia, Magdalena Götz, Rossella Di Giaimo and Silvia Cappello

Review timeline:

Submission date:	9th Sep 2019
Editorial Decision:	14th Oct 2019
Revision received:	12th Feb 2020
Editorial Decision:	9th Mar 2020
Revision received:	1st Apr 2020
Accepted:	2nd Apr 2020

Editor: Celine Carret

Transaction Report:

1st Editorial Decision

14th Oct 2019

Thank you for the submission of your manuscript to EMBO Molecular Medicine. We have now heard back from the three referees whom we asked to evaluate your manuscript.

You will see that while the three referees find the data of interest, they also have similar and overlapping concerns. Of critical relevance, you should provide more supportive mechanism. Additionally, you should rewrite and re-organise the study, in terms of presentation as well and expand the discussion. All three referees commented on this and we believe that the data should be presented and discussed in a clearer and less ambiguous manner.

We would therefore welcome the submission of a revised version within three months for further consideration and would like to encourage you to address all the criticisms raised as suggested to improve conclusiveness and clarity. Please note that EMBO Molecular Medicine strongly supports a single round of revision and that, as acceptance or rejection of the manuscript will depend on another round of review, your responses should be as complete as possible.

***** Reviewer's comments *****

Referee #1 (Remarks for Author):

The manuscript by Di Matteo et al. investigates the role of cystatin B (CSTB) in the early developing brain using mouse and human brain organoids. They found that CSTB can affect the proliferation of the neuronal progenitors and affect the migration of interneurons in a non-cell autonomous manner. The protein is secreted into the mouse cerebral spinal fluid and conditioned medium from human cortical organoids (hCO). This gene is of interest since autosomal-recessive mutations in this gene result in a neurodegenerative disorder known as progressive myoclonus epilepsy (PME) of Unverricht-Lundborg-type (EPM1). The manuscript is of interest and contributes to our overall understanding of different molecular mechanisms regulating brain development.

The authors used either overexpression of wild-type or mutated CSTB by in utero electroporation of mouse embryos or overexpression of CSTB in hCO as well as using hCO derived from patients. The mouse experiments do not directly correspond with the human disease, and their results imply that the mutation can act in a dominant-negative mechanism, which was not clearly discussed.

Specific comments:

CSTB expression during cortical development in hCOs and mice

Figure 1A: I am not really sure what is the significance of the later time points, it looks that the tissue at 85 days is somewhat disintegrating.

1C: No significant differences in the expression of CSTB in PAX6+ versus NEUN+

1E: Please specify what are the additional high expression areas?

The authors indicate that overexpression of CSTB affects the proliferation of progenitors in a non-cell autonomous way. The non-cell autonomous mechanism was tested by quantifying the number of GFP+ cells and pH3+ cells. Yet, what was not accounted for is that the GFP is diluted upon cell division, and cells that did contain GFP will be scored negative following cell division. To test this hypothesis directly, it will be useful either to lineage label the transfected cells, or at least to perform threshold analysis on the GFP+ cells. It is possible that the effect is both cell-autonomous and non-cell autonomous. Some clue for this may be seen in examining the in utero electroporated side and the collateral side. The effect is clear in the side that was electroporated. Since the overexpressed protein is secreted to the CSF, it could be expected that the other side will be affected as well. How CSTB affects cell proliferation is unclear. A more detailed analysis of cell cycle progression is needed. The authors show more Tbr2+ cells, yet did not examine in parallel Pax6+ cells or double positive Tbr2+/Pax6+ cells.

Figure EV2 K: The authors indicate that there is no change in the number of interneurons, yet the number of transfected (red) cells is minute. Even with this small number of cells there is a slight increase, yet not significant.

Figure 2L,M; the number of transfected cells and their distribution is very different, is there a problem with radial migration of the CSTB transfected neurons?

There is also a very striking difference between the GAD65 and the GAD67 experiments, is there an explanation?

Figure 3, please include and compare with GFP only, one- or two-way ANOVA will be needed for measuring statistical significance.

Proliferation is reduced in EPM1-derived cerebral organoids

A more detailed analysis regarding the effect on cell proliferation is required.

If the mutations result in loss of function, is the addition of recombinant protein to the medium of the cerebral human organoids sufficient to rescue the phenotype?

The discussion is very short.

Referee #2 (Comments on Novelty/Model System for Author):

The report is based on the comparison of animal models (induced by in utero electroporation) and brain organoids derived from patients. Both techniques are on the state of the art and particularly suitable for evaluating cortical development. The author greatly contributed to the development of the later mentioned approach.

The project originality is not only based on the type of preparations, but in the notion that phenotype would result from a non-cell autonomous effect of CSTB after its secretion to the extracellular space. However, I do not think that authors reached a fully demonstration of this and additional experiments seem to me necessary for reaching a solid conclusion. I think that brain organoids will give opportunities to evaluate this more in depth.

The quality of the work is optimal, but the presentation is a bit frustrating, disorganized. This can be easily improved (I did comments to authors in this sense).

Data obtained tend to suggest that Progressive myoclonus epilepsy (PME) of Unverricht-Lundborg-type (EPM1) would be a developmental pathology or at least partially linked to developmental alterations. However it is unclear yet if the changes associated to CSTB mutation are really involved on the clinical manifestations and evolution of the disease

Referee #2 (Remarks for Author):

The report of Di Matteo et al. aims to elucidate the role of CSTB, an inhibitor of the cathepsin family of proteases, in developing brain cortex and to provide some hints on the pathophysiological basis of brain disorders (EPM1) linked to mutations of this inhibitor.

To these aims the report analyze first the expression of CSTB in mice embryonic brain and in human cerebral organoids (hCOs). Their data focus on the expression of CSTB in cortical progenitors. These results lead authors to evaluate the involvement of CSTB on cortical proliferation, by inducing the overexpression of CSTB in embryonic mice and hCOs by electroporation and observed intriguing data suggesting non-cell autonomous effects induced by the overexpression of CSTB: expansion of cortical progenitors (TBR2+) and accumulation of interneurons in the electroporated area.

Authors propose that CSTB non-cell autonomous effects on cortical progenitors and interneurons would be the result of a secretion of CSTB to the extracellular space. If their present data and recent publication (Penna et al. 2019) indicate indeed that CSTB can be secreted, they do not provide evidences to confirm that secreted CSTB is responsible for the phenotype evaluated. It is also unclear if in physiological conditions CSTB has a non-cell autonomous effect on cortical progenitors and migrating interneurons. I think that it would be possible to go farther on the demonstration, so nicely initiated by authors, to allow more solid conclusions.

In the second part of the study, authors evaluate the non-cell autonomous effects induced in mice and hCOs by the transfection of a CSTB mutation identified in EPM1 patients and leading to a truncated form of the protein (R68X). Data obtained were compared to samples transfected with the WT form. It is difficult to appreciate from only this comparison if R68X induces a phenotype at all (we lack the comparison with a mismatch construct), though it seems indeed that the phenotype observed after CSTB overexpression is no longer reproduced. Authors also show that the truncated protein is not detected in the conditioned media (degraded? No-detected by the antibodies? No secreted?) suggesting that this is the reason for the differences observed. However, this is still an open question. These data indicate also that the endogenous WT form (produced by transfected and non-transfected cells) is not released in sufficient amounts to be detected and maybe to impact neighbor cells. Is this protein in physiological conditions playing a role on cortical development? Maybe that knockdown experiments would help clarifying this point.

In the third part of the study, authors take advantage of brain organoids generated from EPM1 patients for evaluating developmental pathomechanisms without overexpressing CSTB constructs and in a human relevant condition. Two patients were selected, none involving the same mutation tested in the previous section (it would have been more coherent), but associating an apparent reduction on the expression levels of CSTB transcript and protein as compared to organoids generated from one control individual. Their data indicate that both mutations affect the growing of organoids likely though a reduced proliferation of cortical progenitors. Their data also suggest that mutations paradoxically increased the thickness of the cortical plate, due to an increase on the number of immature neurons DCX+. These data are reminiscent of data obtained in hCOs transfected with R68x but unfortunately the parameters analyzed were not the same and we cannot completely appreciate differences and resemblances.

I find some other points in the manuscript that requires revision:

-figures 2 A-H and EV2 A-H are showing data from the same section but are somehow difficult to read and combine. The main effect observed by authors concern "non-electroporated" progenitor

cells but graphs represent either total cells or GFP+ cells (with the exception of EV2D.) I think that for clarity, figure 2 (main text) should include graphs showing data from both GFP+ and GFP- cells. I do not think that the total population data is helping the understanding of the effects induced by CSTB transfection (a mix of apparently two different effects); they can be moved to EV figure.

- Concerning still this section, I am puzzled that GFP + cells are not affected. If the effect on progenitor cells is due to the secretion of CSBT, it should affect all surrounding cells, including GFP cells, unless they are engaged in a process that alters their proliferation (e.g. apoptosis or differentiation). This is probably a difficult point to resolve, but it is not at the core of the main objective? The elucidation of the role of CSTB on corticogenesis?

-In figure 2F, M and O, quantification by Bin are displayed and authors claim that bin 3 covers the intermediate zone. It will be important for readers to get Bin limits displayed in figures 2L, N and 3G.

-It would help the reader if data are presented along figures following the same model to facilitate comparisons. For example, in Figures 2 and 3, presenting data following the same representation for Tbr2 changes (difficult to compare now 2H and 3H).

- Statistic description did not clarify if authors validated for their statistical comparisons the power strength and consequently confirmed that their sample sizes were adequate. Also, the values of the n in the different experiments were not always clear.

- The discussion contains many vague sentences, lacks precision and does not frankly highlight the topic and the interest of the research. The results obtained must be interpreted first and give substance to hypothesis or nourish controversies, otherwise the work done and the results obtained will remain unnoticed.

Referee #3 (Comments on Novelty/Model System for Author):

The Manuscript by Cappello et al. on "CystatinB is essential for proliferation and interneuron migration in individuals with EPM1 epilepsy" is very interesting as it shows, perhaps for the first time, convincing data on the role of cystatin B in the CNS, starting to explain why CSTB mutants cause such devastating disease. The experiments are well designed and the results consistently point to the strong involvement of this protein in the regulation of the early stages of neurogenesis and in the growth and organization of migrating interneurons in mouse and in human cerebral organoids. In addition the authors show that CSTB is secreted in the conditioned medium of human cerebral organoids as well as in the conditioned medium of E14 mouse cortical cells in culture.

For these reasons I think that these data should be published.

However, the manuscript is very difficult to read, especially in the legend and in the results sections. My suggestion is that the manuscript is revised possibly by a person of English mother language and that the results and legends are presented more clearly. The discussion could also be expanded considering the size of the work.

Referee #3 (Remarks for Author):

1. Technical quality (including statistical analysis).

The data presented are solid, with adequate controls and statistical analysis, use of different techniques, specimens/samples from mouse and human, including EPM1 patients.

2. Strong evidence for the conclusions that are drawn.

All data obtained in a number of different and independent experiments point to the same conclusion: cstb protein is involved both in proliferation of cells in the developing cortex and interneuron migration during CNS neurogenesis. Its overexpression promotes proliferation and thus rises the thickness of the intermediate layer of the cortex. Also interneuron migration of cells close to those that express cstb is positively affected. On the other side, its absence or mutation reduces proliferation and thus the thickness of the intermediate layer of the cortex, the number of migrating

interneurons is diminished as well.

3. Novelty.

For the first time the specific role of *cstb* in CNS, suggested in a number of papers, is shown, enlightening the cause of the primary consequence of its mutation in EPM1 patients: atrophy and reduced thickness of cerebral structures.

4. Medical impact.

First step to understand the molecular basis of EPM1, and thus it opens to further investigations for drug design.

5. Adequacy of model system.

Adequate.

6. Clarity and interest for the non-specialist.

Results are extremely interesting for a wide range of readers, but not always easy to understand, especially for the non-specialist. The reader is requested to constantly check the images to follow the authors' reasoning. The authors should deepen the discussion.

- The Manuscript by Cappello et al. on "CystatinB is essential for proliferation and interneuron migration in individuals with EPM1 epilepsy" is very interesting as it shows, perhaps for the first time, convincing data on the role of cystatin B in the CNS, starting to explain why CSTB mutants cause such devastating disease. The experiments are well designed and the results consistently point to the strong involvement of this protein in the regulation of the early stages of neurogenesis and in the growth and organization of migrating interneurons in mouse and in human cerebral organoids. In addition the authors show that CSTB is secreted in the conditioned medium of human cerebral organoids as well as in the conditioned medium of E14 mouse cortical cells in culture.
- For these reasons I think that these data should be published.
- However, the manuscript is very difficult to read, especially in the legend and in the results sections. My suggestion is that the manuscript is revised possibly by a person of English mother language and that the results and legends are presented more clearly.
- The discussion could also be expanded considering the size of the work. As it stands, it does not underline the relevance, the implication and the novelty of the data.

1st Revision - authors' response

12th Feb 2020

Please see next page.

Referee #1 (Remarks for Author):

The manuscript by Di Matteo et al. investigates the role of cystatin B (CSTB) in the early developing brain using mouse and human brain organoids. They found that CSTB can affect the proliferation of the neuronal progenitors and affect the migration of interneurons in a non-cell autonomous manner. The protein is secreted into the mouse cerebral spinal fluid and conditioned medium from human cortical organoids (hCO). This gene is of interest since autosomal-recessive mutations in this gene result in a neurodegenerative disorder known as progressive myoclonus epilepsy (PME) of Unverricht-Lundborg-type (EPM1). The manuscript is of interest and contributes to our overall understanding of different molecular mechanisms regulating brain development.

The authors used either overexpression of wild-type or mutated CSTB by in utero electroporation of mouse embryos or overexpression of CSTB in hCO as well as using hCO derived from patients. The mouse experiments do not directly correspond with the human disease, and their results imply that the mutation can act in a dominant-negative mechanism, which was not clearly discussed. Specific comments:

CSTB expression during cortical development in hCOs and mice

Figure 1A: I am not really sure what is the significance of the later time points, it looks that the tissue at 85 days is somewhat disintegrating.

We have removed the 85 days as suggested by the reviewer.

1C: No significant differences in the expression of CSTB in PAX6+ versus NEUN+

We have changed our statements regarding CSTB expression according to the reviewer's comment.

1E: Please specify what are the additional high expression areas?

We have added information regarding the specific areas where CSTB is enriched.

The authors indicate that overexpression of CSTB affects the proliferation of progenitors in a non-cell autonomous way. The non-cell autonomous mechanism was tested by quantifying the number of GFP+ cells and pH3+ cells. Yet, what was not accounted for is that the GFP is diluted upon cell division, and cells that did contain GFP will be scored negative following cell division. To test this hypothesis directly, it will be useful either to lineage label the transfected cells, or at least to perform threshold analysis on the GFP+ cells. It is possible that the effect is both cell-autonomous and non-cell autonomous. Some clue for this may be seen in examining the in utero electroporated side and the collateral side. The effect is clear in the side that was electroporated. Since the overexpressed protein is secreted to the CSF, it could be expected that the other side will be affected as well.

We thank the reviewer for this important and relevant comment. The time is quite short (2 days) and we always perform immunostaining with GFP antibody to detect also low level of GFP. But cannot exclude 100% the possibility that the GFP is very quickly diluted and undetectable. Therefore, we invested in an alternative approach, as suggested by the reviewer below. To demonstrate the role of secreted CSTB on proliferation we used the conditioned medium from the control and patient organoids and we switched that every day for a month (Fig 5H and EV5G-J). The surprising and striking result is that we could rescue the proliferation phenotype of the mutant organoids by just providing them the conditioned medium of the control organoids. This result strongly supports the cell non-autonomous mechanism.

How CSTB affects cell proliferation is unclear. A more detailed analysis of cell cycle progression is needed. The authors show more Tbr2+ cells, yet did not examine in parallel Pax6+ cells or double positive Tbr2+/Pax6+ cells.

We appreciate the suggestions of the reviewer and we have performed a more detailed analysis on both specific cell cycle phases and cell fate. Particularly, we investigated the number of cells in G1-phase by quantifying Cyclin D1+ cells, S-phase by injecting (30 minutes) and counting BrdU+ cells and M-phase by Ph3 analysis. The result is that we found more cells in S-phase suggesting a cell cycle specific effect of CSTB on neural progenitors (Fig 2 and EV2).

To evaluate if CSTB has a more prominent role on a specific type of progenitors, we performed Pax6 and Tbr2 immunostainings. Single and double quantifications revealed a similar increase in all 3 populations, suggesting a general function of CSTB on progenitors (Fig 2).

Figure EV2 K: The authors indicate that there is no change in the number of interneurons, yet the number of transfected (red) cells is minute. Even with this small number of cells there is a slight increase, yet not significant.

The reviewer is correct, we do not detect a significant increase in the number of interneurons, as shown in the new Fig EV3 E-F even though the electroporated area is not very small (Fig 3D and 3F). Fig EV3G shows a detail of the analyzed section.

Figure 2L,M; the number of transfected cells and their distribution is very different, is there a problem with radial migration of the CSTB transfected neurons?

We thank the reviewer for spotting this detail. Indeed, the panel representing the experiment using GAD67 was simply a maximum projection of a section. On the contrary the panel representing the experiment using GAD65 is a single confocal optical section. We exchanged the representative picture, now both are single optical sections, to make them comparable as the reviewer suggested (see new Fig 3 D and F).

There is also a very striking difference between the GAD65 and the GAD67 experiments, is there an explanation?

We exchanged the representative picture to make them comparable as the reviewer suggested.

Figure 3, please include and compare with GFP only, one- or two-way ANOVA will be needed for measuring statistical significance.

We added CTRL results in every plot of new Fig 4

Proliferation is reduced in EPM1-derived cerebral organoids. A more detailed analysis regarding the effect on cell proliferation is required.

If the mutations result in loss of function, is the addition of recombinant protein to the medium of the cerebral human organoids sufficient to rescue the phenotype?

We thank the reviewer for this crucial comment. We have performed the experiment suggested and it's now shown in Fig 5H and EV5G-J

The discussion is very short.

We have expanded the discussion as requested.

Referee #2 (Comments on Novelty/Model System for Author):

The report is based on the comparison of animal models (induced by in utero electroporation) and brain organoids derived from patients. Both techniques are on the state of the art and particularly suitable for evaluating cortical development. The author greatly contributed to the development of the later mentioned approach.

The project originality is not only based on the type of preparations, but in the notion that phenotype would result from a non-cell autonomous effect of CSTB after its secretion to the extracellular space. However, I do not think that authors reached a fully demonstration of this and additional experiments seem to me necessary for reaching a solid conclusion. I think that brain organoids will give opportunities to evaluate this more in depth.

We thank the reviewer for the positive comments and feedback on our work.

The quality of the work is optimal, but the presentation is a bit frustrating, disorganized. This can be easily improved (I did comments to authors in this sense).

Data obtained tend to suggest that Progressive myoclonus epilepsy (PME) of Unverricht-Lundborg-type (EPM1) would be a developmental pathology or at least partially linked to developmental alterations. However it is unclear yet if the changes associated to CSTB mutation are really involved on the clinical manifestations and evolution of the disease.

Referee #2 (Remarks for Author):

The report of Di Matteo et al. aims to elucidate the role of CSTB, an inhibitor of the cathepsin family

of proteases, in developing brain cortex and to provide some hints on the pathophysiological basis of brain disorders (EPM1) linked to mutations of this inhibitor.

To these aims the report analyze first the expression of CSTB in mice embryonic brain and in human cerebral organoids (hCOs). Their data focus on the expression of CSTB in cortical progenitors. These results lead authors to evaluate the involvement of CSTB on cortical proliferation, by inducing the overexpression of CSTB in embryonic mice and hCOs by electroporation and observed intriguing data suggesting non-cell autonomous effects induced by the overexpression of CSTB: expansion of cortical progenitors (TBR2+) and accumulation of interneurons in the electroporated area.

Authors propose that CSTB non-cell autonomous effects on cortical progenitors and interneurons would be the result of a secretion of CSTB to the extracellular space. If their present data and recent publication (Penna et al. 2019) indicate indeed that CSTB can be secreted, they do not provide evidences to confirm that secreted CSTB is responsible for the phenotype evaluated. It is also unclear if in physiological conditions CSTB has a non-cell autonomous effect on cortical progenitors and migrating interneurons. I think that it would be possible to go farther on the demonstration, so nicely initiated by authors, to allow more solid conclusions.

We thank the reviewer for this comment. It's definitely a crucial point of this study and we therefore invested in alternative approaches and perform a full set of new experiments. To demonstrate the role of secreted CSTB on proliferation we used the conditioned medium from the control and patient organoids and we switched that every day for a month (Fig 5H and EV5G-J). The surprising and striking result is that we could rescue the proliferation phenotype of the mutant organoids but just providing them the conditioned medium of the control organoids. Moreover, to investigate possible mechanisms contributing to the dysfunction of CSTB in patients, we have performed proteomic analysis on patient's hCOs (Fig 6 and EV6). The results show a clear rearrangement in the extracellular matrix and secretory pathways, possibly contributing to the cell non-autonomous phenotype.

In the second part of the study, authors evaluate the non-cell autonomous effects induced in mice and hCOs by the transfection of a CSTB mutation identified in EPM1 patients and leading to a truncated form of the protein (R68X). Data obtained were compared to samples transfected with the WT form. It is difficult to appreciate from only this comparison if R68X induces a phenotype at all (we lack the comparison with a mismatch construct), though it seems indeed that the phenotype observed after CSTB overexpression is no longer reproduced. Authors also show that the truncated protein is not detected in the conditioned media (degraded? No-detected by the antibodies? No secreted?) suggesting that this is the reason for the differences observed. However, this is still an open question. These data indicate also that the endogenous WT form (produced by transfected and non-transfected cells) is not released in sufficient amounts to be detected and maybe to impact neighbor cells. Is this protein in physiological conditions playing a role on cortical development? Maybe that knockdown experiments would help clarifying this point.

This is also a very relevant comment as we want to understand and investigate if the malfunction of the R68X is only a null function or if this R68X has a dominant effect. We therefore performed knockdown experiments using miRNA targeting CSTB. In the new Fig 4 of the manuscript we present and compare the different conditions: control, knockdown, overexpression of wt and of pathological CSTB. We found that the pathological R68X has an effect that resembles the knockdown, suggesting a dominant effect. As suggested by the reviewer we now show side by side all the quantifications to help the readers to compare between conditions. We also included a WB of the protein extracts from mouse primary cells (Fig EV4D) to show that the absence of R68X signal in WB in Fig EV4C is not due to detection problems of the antibody.

In the third part of the study, authors take advantage of brain organoids generated from EPM1 patients for evaluating developmental pathomechanisms without overexpressing CSTB constructs and in a human relevant condition. Two patients were selected, none involving the same mutation tested in the previous section (it would have been more coherent), but associating an apparent reduction on the expression levels of CSTB transcript and protein as compared to organoids generated from one control individual. Their data indicate that both mutations affect the growing of organoids likely though a reduced proliferation of cortical progenitors. Their data also suggest that mutations paradoxically increased the thickness of the cortical plate, due to an increase on the number of

immature neurons DCX+. These data are reminiscent of data obtained in hCOs transfected with R68x but unfortunately the parameters analyzed were not the same and we cannot completely appreciate differences and resemblances.

As the reviewer suggests, generation of hCOs from patients with R68X mutation would be more coherent. Our hope was to be able to compare both type of mutations, targeting the promoter region and the coding region. We collected PBMCs from 2 R68X patients but, despite several attempts, we were unable to reprogram the PBMCs and generate iPSCs. We hypothesized that this is due to the dominant effect causing premature differentiation. Therefore we decided to use 2 EPM1 patients with different genetic background.

The analysis of the R68X electroporated was mainly focusing on the proliferation to mimic the results obtained in the mouse cortex and therefore we only analyzed early timepoints. For a more detailed analysis in organoids, we preferred to switch to patients' organoids, to be able to mimic as close as possible the *in vivo* conditions, where the mutation is present from the beginning and in all cells.

I find some other points in the manuscript that requires revision:

-figures 2 A-H and EV2 A-H are showing data from the same section but are somehow difficult to read and combine. The main effect observed by authors concern "non-electroporated" progenitor cells but graphs represent either total cells or GFP+ cells (with the exception of EV2D.) I think that for clarity, figure 2 (main text) should include graphs showing data from both GFP+ and GFP- cells. I do not think that the total population data is helping the understanding of the effects induced by CSTB transfection (a mix of apparently two different effects); they can be moved to EV figure.

We reported the cell proliferation of electroporated cells by counting Ki67+GFP+ in FIG EV2A, EV2B and EV4B for OE and KD of CSTB and we did not find any significant change, therefore we always focused on the total population. Only upon OE of R68X in Fig 4G, we detect a significant increase in the cell-autonomous phenotype that we further investigated for premature differentiation in Fig EV4E-F comparing the 3 conditions OE, KD and R68X OE.

- Concerning still this section, I am puzzled that GFP + cells are not affected. If the effect on progenitor cells is due to the secretion of CSBT, it should affect all surrounding cells, including GFP cells, unless they are engaged in a process that alters their proliferation (e.g. apoptosis or differentiation). This is probably a difficult point to resolve, but it is not at the core of the main objective? The elucidation of the role of CSTB on corticogenesis?

We agree with the reviewer but it's very difficult to find striking effects as the phenotype is more or less pronounce depending on the amount of electroporated cells.

-In figure 2F, M and O, quantification by Bin are displayed and authors claim that bin 3 covers the intermediate zone. It will be important for readers to get Bin limits displayed in figures 2L, N and 3G.

We thank the reviewer for the comment. However, overlapping the bins on the picture results in difficulties in visualizing the electroporated/immunostained cells and the bins are just an approximation of the zones (Ventricular Zone, Subventricular Zone, Intermediate Zone and Cortical Plate). Importantly we applied the same binning criteria to each picture, as the binning has always been performed by subdividing the cerebral cortex in 5 equally-distributed bins as clarified in the text.

-It would help the reader if data are presented along figures following the same model to facilitate comparisons. For example, in Figures 2 and 3, presenting data following the same representation for Tbr2 changes (difficult to compare now 2H and 3H).

We thank the reviewer for this useful comment. We adjusted the figures accordingly.

- Statistic description did not clarify if authors validated for their statistical comparisons the power strength and consequently confirmed that their sample sizes were adequate. Also, the values of the n in the different experiments were not always clear.

We thank the reviewer for this useful comment. We clarified what is referred as n in every figure legend

- The discussion contains many vague sentences, lacks precision and does not frankly highlight the topic and the interest of the research. The results obtained must be interpreted first and give substance to hypothesis or nourish controversies, otherwise the work done and the results obtained will remain unnoticed.

We have expanded and revised the discussion for clarity.

Referee #3 (Comments on Novelty/Model System for Author):

The Manuscript by Cappello et al. on "CystatinB is essential for proliferation and interneuron migration in individuals with EPM1 epilepsy" is very interesting as it shows, perhaps for the first time, convincing data on the role of cystatin B in the CNS, starting to explain why CSTB mutants cause such devastating disease. The experiments are well designed and the results consistently point to the strong involvement of this protein in the regulation of the early stages of neurogenesis and in the growth and organization of migrating interneurons in mouse and in human cerebral organoids. In addition the authors show that CSTB is secreted in the conditioned medium of human cerebral organoids as well as in the conditioned medium of E14 mouse cortical cells in culture.

For these reasons I think that these data should be published.

However, the manuscript is very difficult to read, especially in the legend and in the results sections. My suggestion is that the manuscript is revised possibly by a person of english mother language and that the results and legends are presented more clearly. The discussion could also be expanded considering the size of the work.

We thank the reviewer for his positive comments and suggestions. We have heavily revised the manuscript accordingly to her/his comments

Referee #3 (Remarks for Author):

1. Technical quality (including statistical analysis).

The data presented are solid, with adequate controls and statistical analysis, use of different techniques, specimens/samples from mouse and human, including EPM1 patients.

2. Strong evidence for the conclusions that are drawn.

All data, obtained in a number of different and independent experiments point to the same conclusion: cstb protein is involved both in proliferation of cells in the developing cortex and interneuron migration during CNS neurogenesis. Its overexpression promotes proliferation and thus rises the thickness of the intermediate layer of the cortex. Also interneuron migration of cells close to those that express cstb is positively affected. On the other side, its absence or mutation reduces proliferation and thus the thickness of the intermediate layer of the cortex, the number of migrating interneurons is diminished as well.

3. Novelty.

For the first time the specific role of cstb in CNS, suggested in a number of papers, is shown, enlightening the cause of the primary consequence of its mutation in EPM1 patients: atrophy and reduced thickness of cerebral structures.

4. Medical impact.

First step to understand the molecular basis of EPM1, and thus it opens to further investigations for drug design.

5. Adequacy of model system.

Adequate.

6. Clarity and interest for the non-specialist.

Results are extremely interesting for a wide range of readers, but not always easy to understand, especially for the non-specialist. The reader is requested to constantly check the images to follow the authors' reasoning. The authors should deepen the discussion.

We thank the reviewer for comments and suggestions, we expanded our discussion accordingly.

- The Manuscript by Cappello et al. on "CystatinB is essential for proliferation and interneuron migration in individuals with EPM1 epilepsy" is very interesting as it shows, perhaps for the first time, convincing data on the role of cystatin B in the CNS, starting to explain why CSTB mutants cause such devastating disease. The experiments are well designed and the results consistently point to the strong involvement of this protein in the regulation of the early stages of neurogenesis and in the growth and

organization of migrating interneurons in mouse and in human cerebral organoids. In addition the authors show that CSTB is secreted in the conditioned medium of human cerebral organoids as well as in the conditioned medium of E14 mouse cortical cells in culture.

- For these reasons I think that these data should be published.
- However, the manuscript is very difficult to read, especially in the legend and in the results sections. My suggestion is that the manuscript is revised possibly by a person of english mother language and that the results and legends are presented more clearly.
- The discussion could also be expanded considering the size of the work. As it stands, it does not underline the relevance , the implication and the novelty of the data.

We thank the reviewer for carefully suggesting where the manuscript can be improved. We re-wrote the manuscript accordingly.

Thank you for the submission of your revised manuscript to EMBO Molecular Medicine. We have now received the enclosed reports from the referees that were asked to re-assess it. As you will see the reviewers are now globally supportive and I am pleased to inform you that we will be able to accept your manuscript pending the following final amendments:

1) Please address the minor text change commented by referee 2.

Please address both referees' comments in writing.

Please provide a point-by-point letter INCLUDING my comments as well as the reviewer's reports and your detailed responses to their comments (as Word file).

***** Reviewer's comments *****

Referee #1 (Comments on Novelty/Model System for Author):

The models used in this study included state of the art technique and are suitable to investigate the scientific question.

Referee #1 (Remarks for Author):

The authors have responded to previous critics in a suitable manner.

Referee #2 (Comments on Novelty/Model System for Author):

Quality and adequacy of model: Techniques are on the state of the art, with strong data obtained with human brain organoids particularly well adapted to the topic investigated. Control experiments well designed

Novelty : demonstration that Cstb plays an important regulatory role on cortical development, is totally new and important to know by the community

Medical impact is very good, as provides a rationale for explaining eventually some clinical manifestations but data won't translate to therapy or diagnosis

Referee #2 (Remarks for Author):

The manuscript has been considerably remodeled and modified to solve the major criticisms. Importantly, authors performed new experiments to evaluate the impact of Cstb downregulation in cortical progenitors and to evaluate whether Cstb effects are mediated by non-cell autonomous mechanisms. Altogether these changes provide a more convincing demonstration that in developing cortex Cstb modulates proliferation of cortical progenitors and influences the migration of interneurons.

There are still a few points that require revision:

Last sentences of abstract and introduction seem a bit strong to me. Proteomic analysis revealed indeed changes affecting mainly extracellular matrix proteins, but in the absence of additional data, validations, etc. I would downscale the notion indicating that these data suggest....

Page 5 the sentence "Taken together, our results.... and neuronal." Seems to be truncated.

In the result section references to figures is sometimes erroneous. For example, in page 6, when describing DCX+ neurons, authors cite Fig EVB4C-D instead of EVC4E-F, or when they indicate that R68X overexpression interfered with interneuron migration authors referred to Fig 3I-J, but those figures do not exist. Also, in page 7 authors mentioned Fig 6F instead of 6H and Figure 6G instead of 6L. In page 8 it is mentioned that Fig6I show 178 differentially represented proteins, but the figure only shows GFP /DAPI staining of organoids.

In page 7, the description of the changes observed with DCX immunostaining is a bit misleading. DCX is not a marker of the cortical plate, and it is unclear if all neurons are DCX+ in the preparation. I would be more precise and factual indicating just that DCX immunopositive cells form a wider band in the presumptive cortical plate in UL1 and U14 as compared with controls.

Lettering of figure parts are sometimes weird or follows a rather random attribution, particularly:

-Figure 2, lettering A to K is really disorganized

-Figure 6, F-G are below H-I

-Figure EV5 has J but not I

-Note also that Figure 3E display "control" for any bin

Descriptions of figure parts are wrongly attributed in legends of figures 2 and 6

2nd Revision - authors' response

1st Apr 2020

The authors performed the requested editorial changes.

YOU MUST COMPLETE ALL CELLS WITH A PINK BACKGROUND ↓
PLEASE NOTE THAT THIS CHECKLIST WILL BE PUBLISHED ALONGSIDE YOUR PAPER

Corresponding Author Name: Silvia Cappello and Rossella Di Giaimo

Journal Submitted to: EMBO molecular medicine

Manuscript Number: EMM-2019-11419-V3